# Combination of ethyl acetate fraction from *Calotropis gigantea* stem bark and sorafenib induces apoptosis in HepG2 cells

**Pattaraporn Chaisupasakul[1,2], Dumrongsak Pekthong[2,3,4], Apirath Wangteeraprasert[5], Worasak Kaewkong[6], Julintorn Somran[7], Naphat Kaewpaeng[2,8], Supawadee Parhira[2,4,9☯ *], Piyarat Srisawang[1,2,10☯ *]**

1 Department of Physiology, Faculty of Medical Science, Naresuan University, Phitsanulok, Thailand, 2 Center of Excellence for Innovation in Chemistry, Naresuan University, Phitsanulok, Thailand, 3 Department of Pharmacy Practice, Faculty of Pharmaceutical Sciences, Naresuan University, Phitsanulok, Thailand, 4 Center of Excellence for Environmental Health and Toxicology, Faculty of Pharmaceutical Sciences, Naresuan University, Phitsanulok, Thailand, 5 Department of Medicine, Faculty of Medicine, Naresuan University, Phitsanulok, Thailand, 6 Department of Biochemistry, Faculty of Medical Science, Naresuan University, Phitsanulok, Thailand, 7 Department of Pathology, Faculty of Medicine, Naresuan University, Phitsanulok, Thailand, 8 Department of Pharmaceutical Chemistry and Pharmacognosy, Faculty of Pharmaceutical Sciences, Naresuan University, Phitsanulok, Thailand, 9 Department of Pharmaceutical Technology, Faculty of Pharmaceutical Sciences, Naresuan University, Phitsanulok, Thailand, 10 Center of Excellence in Medical Biotechnology, Faculty of Medical Science, Naresuan University, Phitsanulok, Thailand

☯ These authors contributed equally to this work.
* supawadeep@nu.ac.th (SP); piyarats@nu.ac.th (PS)

**Data Availability Statement:** We have used the public repository "Zenodo" to deposit the

## Abstract

The cytotoxicity of the ethyl acetate fraction of the *Calotropis gigantea* (L.) Dryand. (*C. gigantea*) stem bark extract (CGEtOAc) has been demonstrated in many types of cancers. This study examined the improved cancer therapeutic activity of sorafenib when combined with CGEtOAc in HepG2 cells. The cell viability and cell migration assays were applied in HepG2 cells treated with varying concentrations of CGEtOAc, sorafenib, and their combination. Flow cytometry was used to determine apoptosis, which corresponded with a decline in mitochondrial membrane potential and activation of DNA fragmentation. Reactive oxygen species (ROS) levels were assessed in combination with the expression of the phosphatidylinositol-3-kinase (PI3K)/ protein kinase B (Akt)/ mammalian target of rapamycin (mTOR) pathway, which was suggested for association with ROS-induced apoptosis. Combining CGEtOAc at 400 μg/mL with sorafenib at 4 μM, which were their respective half-$IC_{50}$ concentrations, significantly inhibited HepG2 viability upon 24 h of exposure in comparison with the vehicle and each single treatment. Consequently, CGEtOAc when combined with sorafenib significantly diminished HepG2 migration and induced apoptosis through a mitochondrial-correlation mechanism. ROS production was speculated to be the primary mechanism of stimulating apoptosis in HepG2 cells after exposure to a combination of CGEtOAc and sorafenib, in association with PI3K/Akt/mTOR pathway suppression. Our results present valuable knowledge to support the development of anticancer regimens derived from the CGEtOAc with the chemotherapeutic agent sorafenib, both of which were administered at half-$IC_{50}$, which may minimize the toxic implications of cancer treatments while improving the therapeutic effectiveness toward future medical applications.

Supporting information, S1-S6, (DOI: 10.5281/zenodo.10799946).

**Funding:** SP and PS received grant supported from National Science Research and Innovation Fund (NSRF) of Thailand [Grant NO. R2564B007]. AW received grant supported from National Science Research and Innovation Fund (NSRF) of Thailand [Grant NO. R2564B033]. SP received grant supported from Agricultural Research Development Agency (Public Organization) [Grant NO. CRP6505030030]. PC received grant supported from Center of Excellence for Innovation in Chemistry (PERCH-CIC) [Grant NO. NUPMEM02/63] and Faculty of Medical Science, Naresuan University, Phitsanulok, Thailand [Grant NO. 63064382]. SP, PS and DP received partial support from the Global and Frontier Research University Fund, Naresuan University [Grant NO. R2567C003]. The funders had no role in study design, data collection and analysis, decision to publish, or preparation of the manuscript.

**Competing interests:** The authors have declared that no competing interests exist.

## Introduction

The World Health Organization estimates that by 2040, hepatocellular carcinoma (HCC), which contributes 75–85% of the overall incidence of primary carcinoma of the liver, will cause the deaths of more than 1,300,000 individuals [1]. Sorafenib is commonly administered and considered to be an effective treatment for HCC due to its efficiency in generating a survival advantage in patients with advanced HCC, which cannot be treated by liver surgery [2]. Sorafenib inhibits the activity of RAF proto-oncogene serine/threonine-protein kinase, B-Raf proto-oncogene, serine/threonine kinase, and kinases in the Rat sarcoma virus/Rapidly accelerated fibrosarcoma (Raf)/ Mitogen-activated protein kinase kinase (MEK)/Extracellular signal-regulated kinase (ERK) signaling pathway, resulting in decreased tumor cell proliferation. Sorafenib can also interfere with angiogenesis by suppressing the hepatocyte factor receptor, Fms-like tyrosine kinase, vascular endothelial growth factor receptor-2/-3, platelet-derived growth factor receptor and other tyrosine kinases [3]. However, numerous side effects of sorafenib have been noted including, hypertension, hand-foot skin reaction, diarrhea, fatigue, weight loss, alopecia, anorexia, and vocal abnormalities [1, 4].

The anticancer-promoting effect of sorafenib against Bel7402 HCC cells is accompanied by a rise in reactive oxygen species (ROS) levels. This increases the protein expression levels of Bax, caspase-3, and caspase-9 while decreasing those of Bcl-2 [5]. Sorafenib triggers apoptosis in human liver cancer cells by suppressing the signaling pathways, phosphatidylinositol-3-kinase (PI3K), protein kinase B (Akt), mammalian target of rapamycin (mTOR), and ERK [6], resulting in the activation of Bax and suppression of Bcl-2, which induces caspase-3 and Poly (ADP-ribose) polymerase expression in LM3 HCC cells [7].

The anticancer efficacy of sorafenib is enhanced when combined with other anticancer drugs compared to single therapy. Apoptosis in LM3 HCC cells was enhanced when sorafenib was combined with capsaicin [7]. In combination, sorafenib and bufalin promoted apoptosis in the cells of non-small cell lung cancer NCI-H292 by enhancing ROS production and suppressing mitochondrial membrane potential (MMP) [8]. Sorafenib combined with celastrol upregulated apoptosis in HepG2 and Hep1-6 HCC cells [9]. Similarly, silibinin combined with sorafenib increased apoptosis in HCC Bel-7402 cells via the activation of Akt, ERK, signal transducer and activator of transcription 3 (STAT3), and mitogen-activated protein kinase p38, leading to the activation of cleaved-caspase3 and cleaved-poly (ADP-ribose) polymerase [10]. Thus, combining sorafenib with plant extracts could be an effective form of anticancer therapy.

*Calotropis gigantea* (L.) Dryand (Apocynaceae), widely referred to as a crown flower or gigantic milkweed, appears natively in Africa, Eastern Asia, and Southeast Asia. As an abundant source of cardenolides, it has applications in traditional Chinese and Ayurvedic medical treatments for a wide range of diseases, including cancer [11, 12]. The leaf and root ethanolic extract of *C. gigantea* exhibited cytotoxicity in T47D breast cancer cells [13]. An ethanolic extract of leaves from *C. gigantea* was found to activate cytotoxicity in WiDr human colon cancer cells [14]. The ethanolic extract of the entire *C. gigantea* plant upregulated the expression of the death receptor, Fas, Fas ligand, and caspase-8 in non-small cell lung cancer cells [15]. The leaves and stem methanolic extract of *C. gigantea* triggered apoptosis by elevating the expression of pro-apoptotic proteins in human breast cancer cells [16]. Furthermore, *C. gigantea* leaf dichloromethane and ethyl acetate fractions exhibited cytotoxicity in WiDr human colon cancer cells [14]. Several phytochemicals in *C. gigantea* extracts, including cardenolides, antifeedant nonprotein amino acids, naphthalene, and terpene derivatives, flavonol glycosides, pregnanes, ursanetype triterpenoids, and steroids, have an essential function in the biological properties of the extracts [17]. In addition, the cardiac glycosides, flavonoids, alkaloids,

phenolics, and triterpenoids in *C. gigantea* extract also showed anti-cancer activities in cancer cells [18–24].

According to our previous study, the *C. gigantea* stem bark extracts were abundant with several secondary metabolites including calactin, cardiac glycosides, flavonoids, triterpenoids, phenolics, and alkaloids [25]. The anti-cancer properties of the ethanolic extract of the *C. gigantea* stem bark have been demonstrated, and HCT116 had the lowest $IC_{50}$ value among HT-29 and HepG2 cells [25, 26]. Calotropin, a particular of cardenolide derived from *C. gigantea* stem bark extracts, reportedly exhibits cytotoxicity to enforce malignancies [27].

The ethyl acetate fraction of *C. gigantea* stem bark extracts (CGEtOAc) contains high levels of cardiac glycosides and phenolics but lower levels of triterpenoids, flavonoids, and calotropins than other fractions [25, 26]. Although a monotreatment with CGEtOAc exhibited a less potent anticancer effect in HepG2 cells than in HCT116 cells, studying the enhanced anticancer efficacy of a standard chemotherapeutic treatment when applied in combination with CGEtOAc is important for developing future alternative treatments. To investigate the potential anticancer properties of the *C. gigantea* stem bark extract and the reduction of chemotherapeutic side effects on HepG2 cells, this study examined the combination treatment with a minimized dose of CGEtOAc and sorafenib that demonstrated apoptotic activity mechanisms.

## Materials and methods

### Materials

Hepatocellular carcinoma HepG2 cells (JCRB1054) and normal lung fibroblast IMR-90 cells (JCRB9054) were obtained from the Japanese Collection of Research Bioresources Cell Bank (JCRB Cell Bank), Japan. Dulbecco's Modified Eagle Medium (DMEM) (10-014-CM) was obtained from Corning, USA. Fetal bovine serum (FBS) (16000–044) and penicillin and streptomycin (15140–122) were purchased from Gibco, USA. 3-(4, 5-Dimethylthiazol-2-yl)-2, 5-diphenyltetrazolium bromide (MTT) (D0801) was purchased from Tokyo Chemical Industry CO., LTD., Japan. Anti-PI3 Kinase Antibody, p110β (PI3K) (09482), and Luminata Forte Western HRP Substrate (WBLUF0100) were purchased from Merck, Germany. Annexin V conjugated Alexa Fluor 488 (A13201), Propidium iodide (PI) (P3566), Hoechst 33342 (H1399), 5,5,6,6'-tetrachloro-1,1',3,3' tetraethylbenzimidazoylcarbocyanine iodide (JC-1) (T3168), Horseradish peroxidase-conjugated goat anti-rabbit (65–6120). goat anti-mouse secondary antibody (A28177), 2', 7'-dichlorodihydrofluorescein diacetate (H2DCFDA) (D399), and AKT Pan Monoclonal Antibody (E.32.10) (MA5-14999) were purchased from Invitrogen, USA. H2DCFDA-Cellular ROS Assay Kit (ab113851) and Anti-Mammalian target of rapamycin (mTOR) (ab32028) were purchased from Abcam, USA. β-Actin (8H10D10) Mouse mAb (3700) were purchased from Cell Signaling Technology, USA. E.Z.N.A.Ⓡ Total RNA Kit I (R6834-01) was purchased from Omega Bio-tek, Inc., USA. TetroTM cDNA Synthesis Kit (BIO-65043) was purchased from Meridian Bioscience, USA. 5× HOT FIREPolⓇ EvaGreenⓇ qPCR Mix Plus (no ROX) (08-25-00001) was purchased from Solis Bio-Dyne, Germany. A proteinase inhibitor cocktail (ML051) was purchased from HIMEDIA, India. Mammalian Protein Extraction Reagent (M-PER^TM) (78501) was purchased from Thermo Fisher Scientific, USA.

### Preparation of standardized CGEtOAc

The CGEtOAc was prepared and standardized for calotropin content with high-performance liquid chromatography, as detailed in our previous study [26]. The *C. gigantea* stem bark was harvested, dried at a temperature of 35 ± 7˚C, and ground into powder. The ultrasonic-assisted extraction with 95% ethanol was used to prepare the crude extract before fractionation using

the liquid-liquid chromatography of water and dichloromethane, followed by the rest of the water layer and ethyl acetate. The ethyl acetate fraction was evaporated at 50°C to obtain the dark brown sticky extract. The content of calotropin in CGEtOAc ($2.7 \pm 0.06$ mg calotropin/10 g CGEtOAc) was determined and used as a bioactive marker to standardize the CGEtOAc. The possible m/z of other cardenolides or other compounds found in CGEtOAc are illustrated in S1 Fig. The plant collection followed the national guideline of Thailand for using plants for research (Approval number 0278 under the Plant Varieties Protect Act B.E. 2542 (1999) section 53 from the Department of Agriculture, Ministry of Agricultural and Cooperatives, Thailand). The voucher specimen of dry plant number 005194 was identified in our previous report [26] and preserved for reference in a recognized herbarium in Thailand at PNU herbarium, Faculty of Science, Naresuan University, Phitsanulok, Thailand, 65000.

## Cell culture

Hepatocellular carcinoma HepG2 (JCRB1054) and fibroblast IMR-90 cells (JCRB9054) were cultured in Dulbecco's Modified Eagle Medium containing 10% fetal bovine serum and 1% penicillin and streptomycin under a humidified incubator at 5% $CO_2$ at 37°C. Cells were cultured in a T-25 culture flask using complete media and subcultured when they reached 80–90% confluency every 4 days during the incubation cycle. Cell count was monitored during each subculture to measure the consistency of growth rates before proceeding with the cell treatment procedure. The human cell culture research was approved by the Naresuan University Institutional Review Board (NU-IRB) in Panel 1: Health Sciences, with the approval number P1-0166/2565.

## MTT assay evaluation of cell viability

After incubating HepG2 cells at a density of $1.5 \times 10^4$ cells/well/150 μL in a 96-well plate for 24 h and exposing them to the extracts for an additional 24 h, 3-(4,5-Dimethylthiazol-2-yl)-2,5-diphenyltetrazolium bromide (MTT) was applied at a concentration of 2 mg/mL to assess viable cells based on mitochondrial reductase activity. We then measured absorbance at 595 nm using dimethyl sulfoxide (DMSO) to dissolve the formazan crystals produced by MTT, employing a microplate reader (SpectraMax ABS, Molecular Devices, USA). The $IC_{50}$ values were calculated using GraphPad Prism version 9.

## Analysis of cell migration activity using wound healing technique

After incubating HepG2 cells at a density of $4 \times 10^5$ cells/well/1 mL in a 12-well plate and allowing them to reach 80–90% confluency within 48 h, we induced a cell monolayer to form physical wounds by scratching the attached cells in a continuous straight line at the center of each well, using a 200 μL sterile tip. The scratch wounds were consistent in size in each well to avoid variations between conditions [28–31]. We removed debris and non-attached cells before subsequently exposing the cells to the extracts, sorafenib, and their combination for 0–72 h.

The gap width of the wound was measured and recorded in three regions of each image between the closest points on both sides of the wound using an inverted microscope (IX71, Olympus Corporation, Japan). The results were analyzed as the percentage of the scratch gap using cellSens Standard Ver.2.3. Triplicate measurements of each well were made to obtain the mean value. The scratch widths of the control and treatment wells were measured and normalized according to their respective values at 0 h. The migrated distance was defined as the following formula:

migrated rate (%) = (width of initial wound—width of remaining wound) / width of initial wound $\times$ 100 [32, 33].

## Determination of apoptosis using flow cytometry

HepG2 cells were incubated at a density of $2 \times 10^5$ cells/1 mL in a 12-well plate for 24 h and exposed them to the extracts for an additional 24 h. The degree of apoptosis was then evaluated after harvesting by double staining with Annexin V conjugated Alexa Fluor 488 at 5 μL (25 μg/mL) in 100 μL of 1x annexin-binding buffer and propidium iodide (PI) at 1 μL of 100 μg/mL, according to the manufacturer's instruction. After a 15-min incubation in the dark, 400 μL of cold annexin-binding buffer was added to the cell suspension and cell apoptosis was measured using flow cytometry.

Annexin V-binding phosphatidylserine results in translocation from the inner plasma membrane to the outside membrane, signaling an early apoptotic stage. PI can enter the cell and bind DNA when the cell membrane becomes permeable. The double labeling of Annexin V and PI reveals late-stage apoptosis. CytoFLEX flow cytometry (CytoFLEX S V2-B2-Y3-R2 Flow Cytometer, Beckman Coulter, USA) and CytExpert Version 2.4.0.28 were used to detect the apoptotic cells.

## Determination of DNA fragmentation by staining with Hoechst 33342

After incubating HepG2 cells at a density of $5 \times 10^5$ cells/2 mL on a glass coverslip in a 35 mm culture dish for 24 h and exposing them to the extracts for an additional 24 h, HepG2 cells were treated with a 4% formaldehyde fixative solution for 15 min. Subsequently, they were incubated with Hoechst 33342 at a concentration of 10 μg/mL for 10 min to identify DNA fragmentation following the manufacturer's instructions. DNA fragmentation was indicated by fluorescence intensity, observed under a fluorescent microscope (BX53F2, Olympus Corporation, Japan).

## Fluorescence labeling with JC-1 to evaluate mitochondrial membrane potential (MMP)

After incubating HepG2 cells at a density of $5 \times 10^5$ cells/2 mL on a glass coverslip in a 35 mm culture dish for 24 h and exposing them to the extracts for an additional 24 h, HepG2 cells were treated with 4% formaldehyde for 15 min before being labeled with 10 μg/mL 5,5,6,6'-tetrachloro-1,1',3,3' tetraethylbenzimidazoylcarbocyanine iodide (JC-1) according to the manufacturer's instructions. Healthy mitochondria show that the cationic dye JC-1 can enter to shift to a dimeric form and display red fluorescence. In the cytoplasm of damaged cells, JC-1 exists as a monomeric structure emitting green fluorescence. Using a fluorescent microscope, the fluorescence intensity of the monomeric and dimeric forms of JC-1 was analyzed (BX53F2, Olympus Corporation, Japan).

## ROS production was assessed using an assay kit and fluorescence imaging

After incubating HepG2 cells at a density of $5 \times 10^5$ cells/2 mL on a glass coverslip in a 35 mm culture dish for 24 h and exposing them to the extracts for an additional 24 h, the cellular ROS production levels were detected in HepG2 cells by applying the H2DCFDA-Cellular ROS Assay Kit according to the manufacturer's instruction. The fluorescence degree of cellular ROS levels was detected with a microplate reader at Ex/Em = 485/535 nm (SpectraMax iD3, Molecular Devices, USA).

Staining with 2',7'-dichlorodihydrofluorescein diacetate (H2DCFDA) was also applied to detect cellular ROS production. HepG2 cells were exposed to H2DCFDA at 20 μM, which is deacetylated by esterase to form 2',7'-H2DCF, which is then oxidized by ROS to form fluorescent 2',7'-dichlorofluorescein and ROS levels was monitored using a fluorescent microscope (BX53F2, Olympus Corporation, Japan) at 40× magnification.

### Real-time quantitative reverse transcription polymerase chain reaction for *PI3K/Akt/mTOR* gene expression

After incubating HepG2 cells at a density of $2 \times 10^6$ cells/4 mL in a 60 mm culture dish for 24 h and exposing them to the extracts for an additional 24 h, total RNA was extracted using E.Z.N.A.® Total RNA Kit I following the manufacturer's instructions. The RNA contents were analyzed at absorbances of 260 and 280 nm using a spectrophotometer (NanoDrop™ One/Onec Microvolume, Thermo Scientific, USA). Tetro™ cDNA Synthesis Kit was used for the complementary DNA (cDNA) synthesis from messenger RNA templates at 5 µg/µL for quantitative reverse transcription-polymerase chain reaction for *PI3K*, *Akt*, and *mTOR* following the manufacturer's instructions. Then, cDNA at 1 ng/µL was used for real-time PCR analysis, which was carried out using 5× HOT FIREPol® EvaGreen® qPCR Mix Plus (no ROX) in a CFX Connect Real-Time PCR System (Bio-rad CFX manager version 3.1) at the following conditions: 40 cycles 95˚C for 30 s; 60˚C for 30 s; 72˚C for 30 s. The relative expression level of genes was quantified and normalized to that of GAPDH as the housekeeping gene using the $2^{-\Delta\Delta Ct}$ method [22]. The sequences of the primers (10 pg/µL) are listed in Table 1.

### Immunoblotting

After incubating HepG2 cells at a density of $2 \times 10^6$ cells/4 mL in a 60 mm culture dish for 24 h and exposing them to the extracts for an additional 24 h, the total cellular protein was extracted with the Mammalian Protein Extraction Reagent, adding 1% proteinase inhibitor cocktail and quantifying with a spectrophotometer (The NanoDrop® ND-1000, Thermo Scientific, USA). Protein samples at 60 µg/µL were separated with 8–12% sodium dodecyl sulfate (SDS)-polyacrylamide gel electrophoresis with 1X running buffer (125 mM Tris-base, 959 mM glycine, 0.1% SDS), transferred to a polyvinylidene fluoride (PVDF) membrane using transfer buffer (25 mM Tris-base, 192 mM glycine, 20% methanol, 0.1% SDS), and nonspecific proteins were blocked with Immobilon® Block-CH (Chemiluminescent Blocker). Membranes were incubated overnight with anti-PI3K (1:1,000), anti-Akt (1:1,000), anti-mTOR (1:1,000) primary antibodies. Then, membranes were incubated for 2–4 h with 1:5,000 concentrations of horseradish peroxidase-conjugated goat anti-rabbit or goat anti-mouse secondary antibody at 4˚C. The expression of β-actin (at 1:1,500 concentration) behaves as a loading control. The 1X phosphate-buffered saline with Tween 20 was used for washing each step of primary and secondary incubation. The image of protein expressions was measured by applying the Luminata Forte Western HRP Substrate and Chemiluminescence western blot detection (Image Quant LAS 4000; GE Healthcare Life Sciences, USA). Protein expression levels relative to β-actin were calculated using ImageJ version 1.46.

### Statistical analysis

The significant differences in data presented as the mean ± SD from at least three different experiments were analyzed with a one-way analysis of variance (ANOVA) using Tukey's honestly significant difference (HSD) test at $p < 0.05$ using GraphPad Prism version 9.

**Table 1. The primer sequences used in RT-qPCR.**

| Gene | Forward primer (5'-3) | Reverse primer (5'-3) |
|------|------------------------|------------------------|
| *GAPDH* | AACGGGAAGCTTGTCATCAATGGAAA | GCATCAGCAGAGGGGGCAGAG |
| *PI3K* | CCACGETOACCCATCATCAGGTGAA | CCTCACGGAGGCATTCTAAAGT |
| *Akt* | CCTCCACGETOACCATCGCACTG | TCACAAAGAGCCCTCCATTATCA |
| *mTOR* | ATGCAGCTGTCCTGGTTCTC | AATCAGACAGGCACGETOACAGGG |

## Results

### Cytotoxicity of CGEtOAc and sorafenib in HepG2 cells

The cytotoxic properties in HepG2 cells assessed using the MTT technique revealed that CGEtOAc, sorafenib, and their combination diminished cell viability in a concentration-responsive manner after 24 h of exposure (Fig 1A–1D). The $IC_{50}$ values for CGEtOAc (0–2,000 μg/mL) and sorafenib (0–40 μM) were 707.87 ± 49.05 μg/mL and 8.65 ± 0.33 μM, respectively. The $IC_{50}$ graphs are presented in S2 Fig. Subsequently, the combination of sub-$IC_{50}$ and around-$IC_{50}$ concentrations of CGEtOAc at 200, 400, 600, and 800 μg/mL and sorafenib at 2, 4, and 8 μM was examined. Fig 1C demonstrates that CGEtOAc at 400 μg/mL in combination with sorafenib at 4 μM significantly decreased cell viability compared to the vehicle and single treatments with CGEtOAc and sorafenib at the same concentration. Furthermore, the cytotoxic effect of this combination was comparable to CGEtOAc at 600 and 800 μg/mL in combination with sorafenib at 4 μM. Notably, the combination of CGEtOAc and sorafenib at an $IC_{50}$ dose of 8 μM was extremely cytotoxic to HepG2 cells.

The combination efficacy of CGEtOAc with sorafenib after 24 h of incubation was analyzed using the combination index (CI) based on the Chou-Talalay method [34–36] using the CompuSyn version 1.0.1 Software (ComboSyn Inc., NJ, USA), as shown in S3 Fig. We found that CGEtOAc at 400 μg/mL in combination with sorafenib at 4 μM had a CI value of 0.5, and the inhibition rate was approximately 60%, indicating the synergistic effect. However, a combination of CGEtOAc at 200 μg/mL and sorafenib at 4 mM also yielded a CI of 0.5, but the inhibition rate was 40%. Although CI values less than 1 were found when combining CGEtOAc at 200, 400, 600, and 800 μg/mL with sorafenib at a high dose of 8 μM, it is important to note that sorafenib was used at a high concentration. Therefore, we concluded that CGEtOAc at 400 μg/mL and sorafenib at 4 μM were suitable for the following experiments.

In addition, the toxicity to IMR-90 cells of CGEtOAc at 200, 400, and 800 μg/mL was significant in comparison to the vehicle, but to a lesser degree than that observed in HepG2 cells, while the toxicity of CGEtOAc at 1,000, 1,500, and 2,000 μg/mL was not significantly different from that of the 800 μg/mL solution (Fig 1E). The $IC_{50}$ of CGEtOAc to IMR-90 cells was approximately greater than 5,000 μg/mL. Compared to the vehicle control, the combination of 400 μg/mL CGEtOAc and 4 μM sorafenib had a significant cytotoxic effect on IMR-90 cells, but to a lesser extent than HepG2 cells, where no combination effect was observed (Fig 1F).

### CGEtOAc and sorafenib, singly and in combination inhibited the migration of HepG2 cells

At a concentration of 400 μg/mL for CGEtOAc, 4 μM for sorafenib, and their combination, cell migration was significantly inhibited. This was demonstrated by a higher percentage of gap distance in a wound healing assay, compared to each hour of incubation in the vehicle group, when 0 h set at 100% for each group (Fig 2A and 2B). The percentage of gap distance for the combination of CGEtOAc and sorafenib in each incubation period did not differ from the single treatments. However, the migration data for the combination of CGEtOAc and sorafenib at 72 h disappeared due to the complete loss of cell viability. Cells detached from the plate, rendering it impossible to measure the gap between cell scratches, attributed to the toxic effects of the combination treatment.

The migration rate, as shown in S4 Fig, confirmed a significant inhibition of HepG2 cell migration rate by CGEtOAc at 400 μg/mL, sorafenib at 4 μM, and their combination compared to each hour of incubation in the vehicle group. The migration rate for the combination of CGEtOAc and sorafenib in each incubation period did not differ from the single treatments.

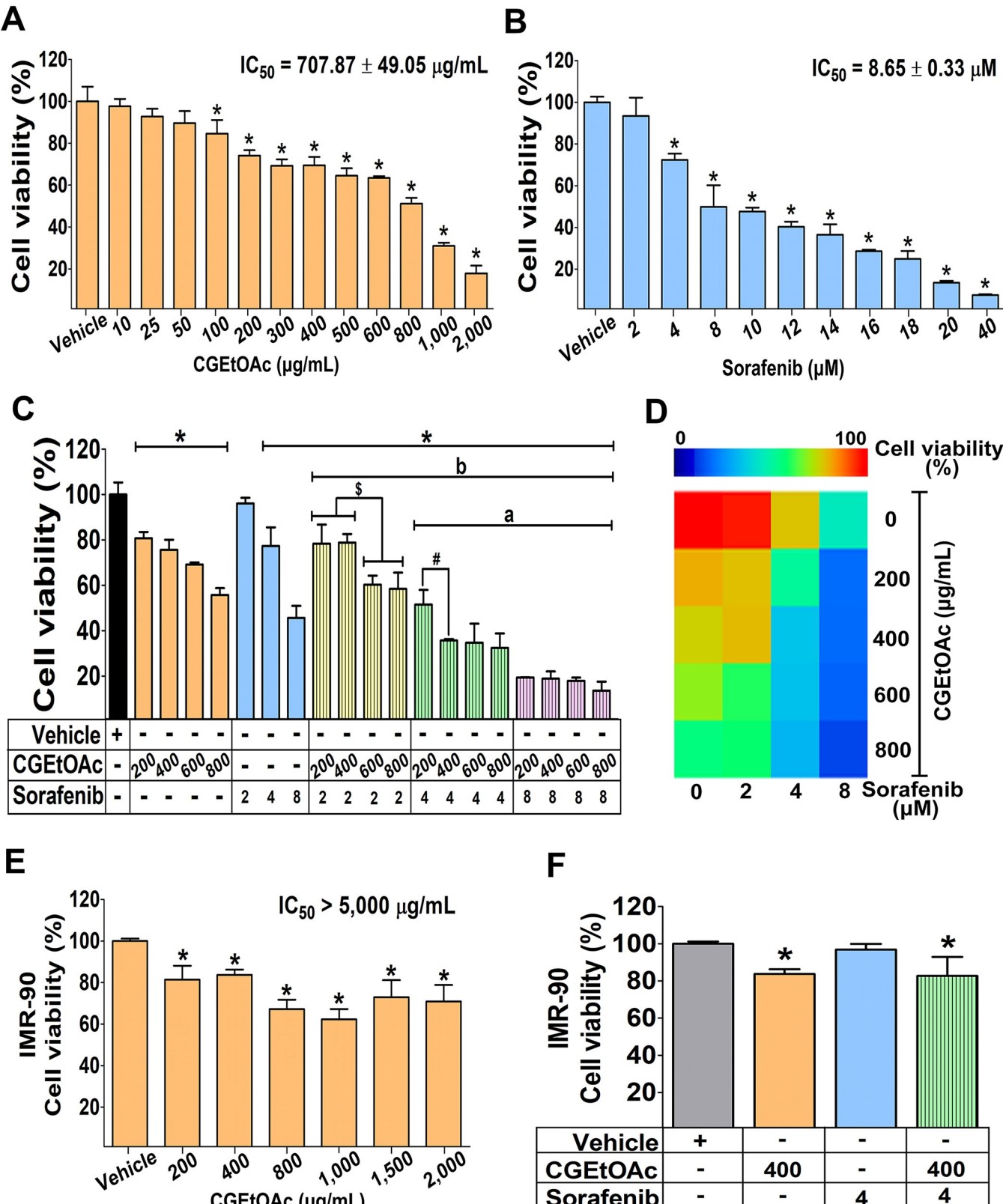

**Fig 1. Cytotoxicity of the ethyl acetate fraction of *C. gigantea* stem bark extracts (CGEtOAc) and sorafenib against HepG2 cells.** The 3-(4, 5-Dimethylthiazol-2-yl)-2, 5-diphenyltetrazolium bromide (MTT) technique was used to assess the cell viability after being treated for 24 h with (A) CGEtOAc,

(B) sorafenib, and (C) their combination demonstrated in the bar graph and (D) heat map analysis. (E) CGEtOAc and (F) the combination of CGEtOAc and sorafenib treatment on IMR-90 cells. Cells treated with 0.8% dimethyl sulfoxide (DMSO) represented the vehicle control. The significant differences in data, presented as the mean ± standard deviation (SD) from at least three different experiments, were investigated with a one-way analysis of variance (ANOVA) using Tukey's honestly significant difference (HSD) test: *; $p < 0.05$ vs the vehicle control, a; $p < 0.05$ vs a single CGEtOAc treatment at their own dose, b; $p < 0.05$ vs a single sorafenib treatment at their own dose.

Our findings indicate that CGEtOAc, sorafenib, and their combination upregulated the anti-proliferative and anti-migratory activities in HepG2 cells.

## Combination of CGEtOAc and sorafenib enhanced apoptosis in HepG2 cells

A combination of CGEtOAc and sorafenib was applied to examine apoptotic HepG2 cells. Annexin V/PI analysis revealed that 400 μg/mL CGEtOAc combined with 4 μM sorafenib significantly increased the proportion of apoptotic cells compared to the vehicle control and single treatments (Fig 3A and 3B), as shown in the gating strategy of flow cytometry demonstrated in S1 Raw images. In addition, JC-1 staining for detecting the dissipation of mitochondrial membrane potential (MMP) was examined. Depolarization of MMP following treatment with 400 μg/mL CGEtOAc combined with 4 μM sorafenib is demonstrated in Fig 4. These results indicate that the promotion of apoptosis in HepG2 cells by the combination of CGEtOAc and sorafenib correlates with MMP damage.

## The combination of CGEtOAc and sorafenib triggered HepG2 apoptosis associated with increasing cellular ROS levels

The underlying mechanism of apoptotic cells in HepG2 was identified by measuring ROS production using H2DCFDA staining. As shown in Fig 5A, 400 μg/mL CGEtOAc combined with 4 μM sorafenib upregulated ROS production with the green fluorescence intensity of dichloro-fluorescein compared to the vehicle control and their respective treatment groups. In addition, Fig 5B demonstrated that the formation of cellular ROS significantly increased following exposure to CGEtOAc, sorafenib, and their combination. ROS levels were approximately 175% and 178% in CGEtOAc and sorafenib treatments, respectively. The treatment with a combination of CGEtOAc and sorafenib significantly increased ROS levels to approximately 193% compared to single treatments. Two hours of pretreatment with N-acetylcysteine (NAC) to inhibit ROS formation confirmed that CGEtOAc, sorafenib, and their combination upregulated ROS generation in HepG2 cells.

Additionally, when ROS generation was inhibited, the apoptotic effect of CGEtOAc, sorafenib, and their combination was considerably reduced but not to baseline levels, in comparison to the vehicle and their single treatment groups (Fig 6A and 6B). However, the downregulation in apoptosis in NAC pretreated at 10 mM for 2 h followed by combined treatment was still significantly greater than in the vehicle and single treatment groups, indicating that ROS is the major regulatory mediator but not the only mechanism mediating the apoptotic effect in HepG2 cells treated with CGEtOAc combined with sorafenib.

## The apoptotic response following treatment with a combination of CGEtOAc and sorafenib may involve the inhibition of the PI3K/Akt/mTOR pathway in HepG2 cells

Apoptosis in cancer cells has been discovered to be mediated by the signaling pathway proteins PI3K/AKT/mTOR [24, 37]. Compared to vehicle control and their respective single treatment

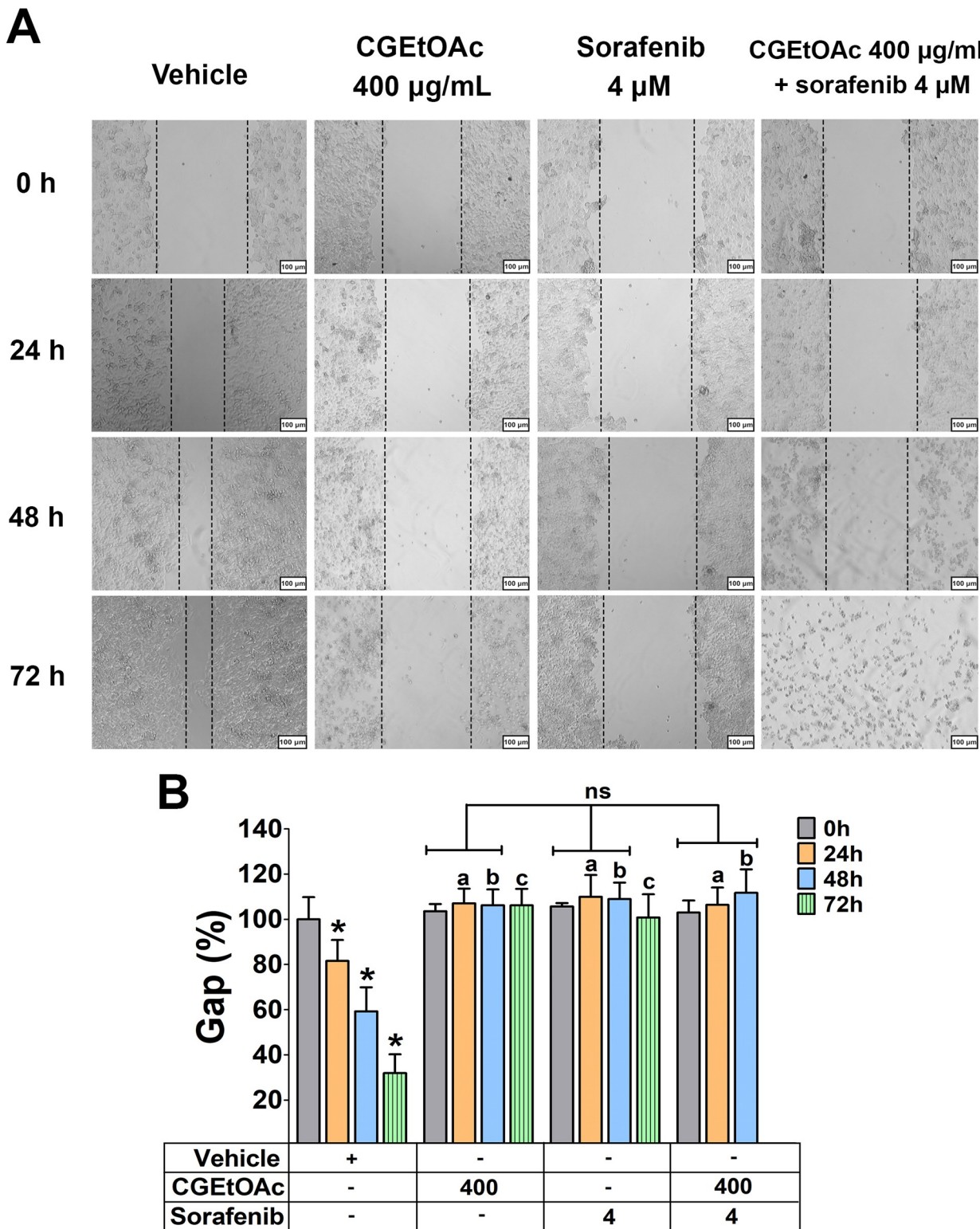

**Fig 2. The migratory capacity of HepG2 cells was assessed through a 0–72 h incubation with CGEtOAc at 400 μg/mL and sorafenib at 4 μM, both singly and in combination.** (A) Wound healing images were captured with a magnification bar of 500 μm. (B) The percentage of the gap was depicted as a bar graph. Cells treated with 0.8% DMSO represented the vehicle control. The significant differences in data, presented as the mean ± SD from at least three different experiments, were investigated with a one-way ANOVA using Tukey's HSD test: *; $p < 0.05$ vs the 0 h of the vehicle control, [a]; $p < 0.05$ compared to 24 h of incubation in the vehicle group, [b]; $p < 0.05$ compared to 48 h of incubation in the vehicle group, and [c]; $p < 0.05$ compared to 72 h of incubation in the vehicle group, with 0 hours set at 100% for each group.

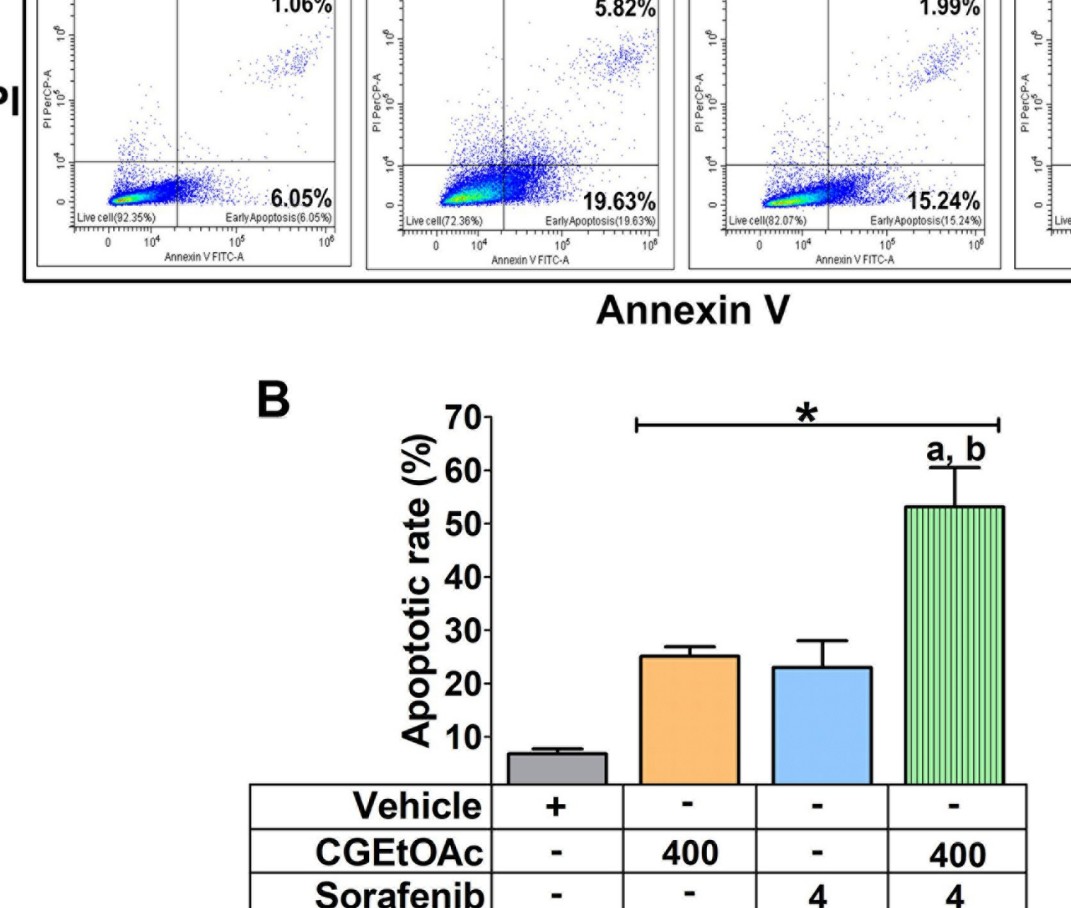

**Fig 3. The apoptotic effect of CGEtOAc, sorafenib, and their combination against HepG2 cells after exposure for 24 h.** (A) Apoptotic levels in HepG2 cells were examined by flow cytometry after double labeling using annexin V and PI. (B) The levels of apoptotic cells were displayed as a bar graph. 0.8% DMSO represented the vehicle control. The significant differences in data, presented as the mean ± SD from at least three different experiments, were investigated with a one-way ANOVA using Tukey's HSD test: *; $p < 0.05$ compared to the vehicle control. [a]; $p < 0.05$ compared to a single CGEtOAc treatment. [b]; $p < 0.05$ compared to a single sorafenib treatment.

groups, 400 μg/mL CGEtOAc combined with 4 μM sorafenib significantly downregulated the PI3K, Akt, and mTOR protein expression. Combined treatment also significantly downregulated the expressions of PI3K, Akt, and mTOR genes (Fig 7). The original uncropped and unadjusted images underlying all blot results are demonstrated in S2 Raw images. Thus, it is hypothesized that the apoptotic mechanism of CGEtOAc combined with sorafenib is associated with the downregulation of the upstream PI3K/AKT/mTOR signaling cascade pathway.

## Discussion

Since 2010, there has been considerable focus on studying the anticancer effects of *C. gigantea* extracts in many cancer models [25, 26, 38–41]. However, biological activities involving *C. gigantea* stem bark extracts have rarely been conducted. The *C. gigantea* stem bark extracts generated cellular toxicity in HCT116, HT-29, and HepG2 cancer cells, of which the

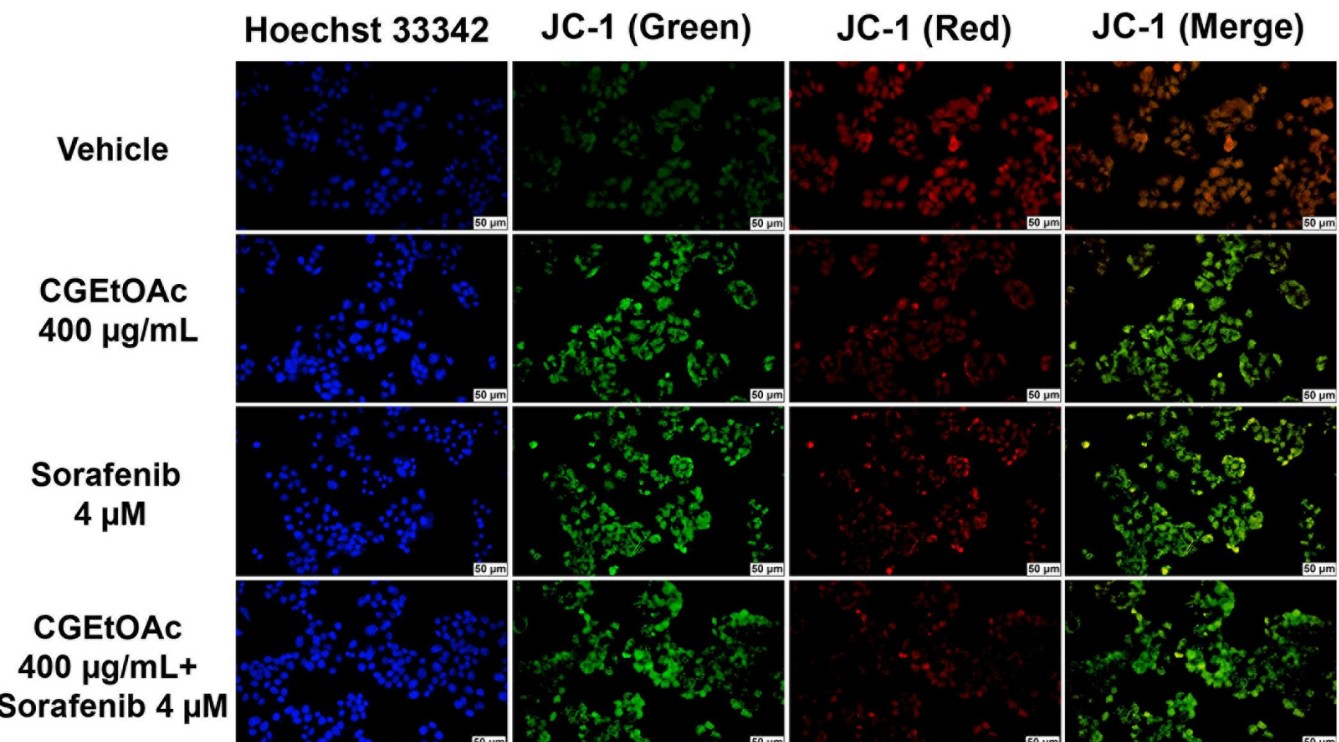

**Fig 4. Determination of mitochondrial membrane potential (MMP) in HepG2 cells after exposure to CGEtOAc, sorafenib, and their combination for 24 h by labeling with 5,5,6,6′-tetrachloro-1,1′,3,3′ tetraethylbenzimidazoylcarbocyanine iodide (JC-1) and Hoechst 33342.** The images were detected using a fluorescent microscope with a magnification bar of 50 μM. Cells treated with 0.8% DMSO represented the vehicle control.

dichloromethane fraction exhibited the most notable apoptotic activity compared to other fractions. HepG2 cell apoptotic responses were much substantially sensitive to each fraction of the extract, and the response to the ethyl acetate fraction treatment was weaker than in HCT116 and HT-29 cells. The ethyl acetate fraction contains the highest levels of cardiac glycosides and phenolics, however, triterpenoids, flavonoids, and calotropins are present in lower levels than dichloromethane and other extract fractions. Nevertheless, this suggests that the cytotoxicity of each fraction from the *C. gigantea* stem bark is not entirely influenced by a single component present in each fraction [25, 26]. Although the overall phytochemical contents and efficacy against HepG2 cells of the ethyl acetate fraction are lower than those of the dichloromethane fraction, including the calotropin content, which was selected as a bioactive marker for standardization [26], the highest amount of total cardiac glycosides in the ethyl acetate fraction is intriguing and merits further investigation when combined with a conventional chemotherapeutic agent, sorafenib at low dose usage, to enhance anticancer efficacy and reduce adverse effects.

Several pieces of evidence suggest that cardiac glycosides derived from plant extracts have a crucial part in the anticancer properties against several cancer cells [42, 43]. Apoptosis in MCF-7 and HepG2 cells was observed by the cardiac glycoside lanatoside C from *Digitalis ferruginea* [22]. Cardenolides, a class of cardiac glycosides also exhibited cytotoxicity to many cancer cells [44, 45]. Consequently, based on the findings in HPLC-EIS-MS, it may be postulated that the cardiac glycoside composition in the ethyl acetate fraction of the *C. gigantea* stem bark extract is one of the key components that confer anticancer properties to the ethyl acetate fraction when combined with sorafenib in HepG2 cells.

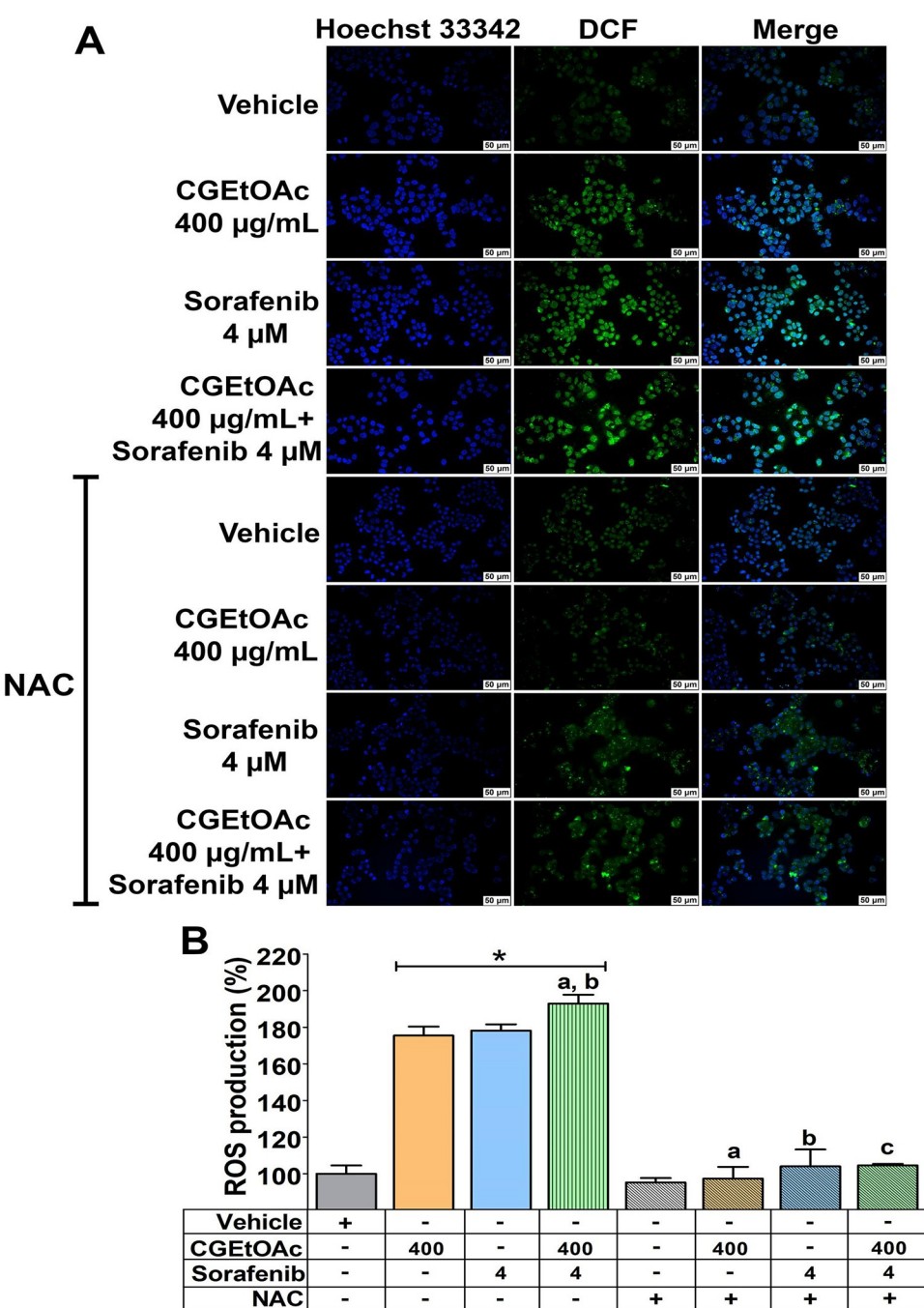

**Fig 5. The intracellular production of reactive oxygen species (ROS) in HepG2 cells subjected to CGEtOAc, sorafenib, and their combination for 24 h.** (A) Cellular ROS levels were analyzed using H2DCFDA fluorescence labeling and visualized under a fluorescence microscope with a magnification bar of 50 μM. (B) Bar graph displaying the percentage of ROS generation. Cells treated with 0.8% DMSO represented the vehicle control. The significant differences in data, presented as the mean ± SD from at least three different experiments, were investigated with a one-way ANOVA using Tukey's HSD test: *; $p < 0.05$ compared to the vehicle control. [a]; $p < 0.05$ compared to a single CGEtOAc treatment. [b]; $p < 0.05$ compared to a single sorafenib treatment. [c]; $p < 0.05$ compared to a combination treatment of CGEtOAc and sorafenib.

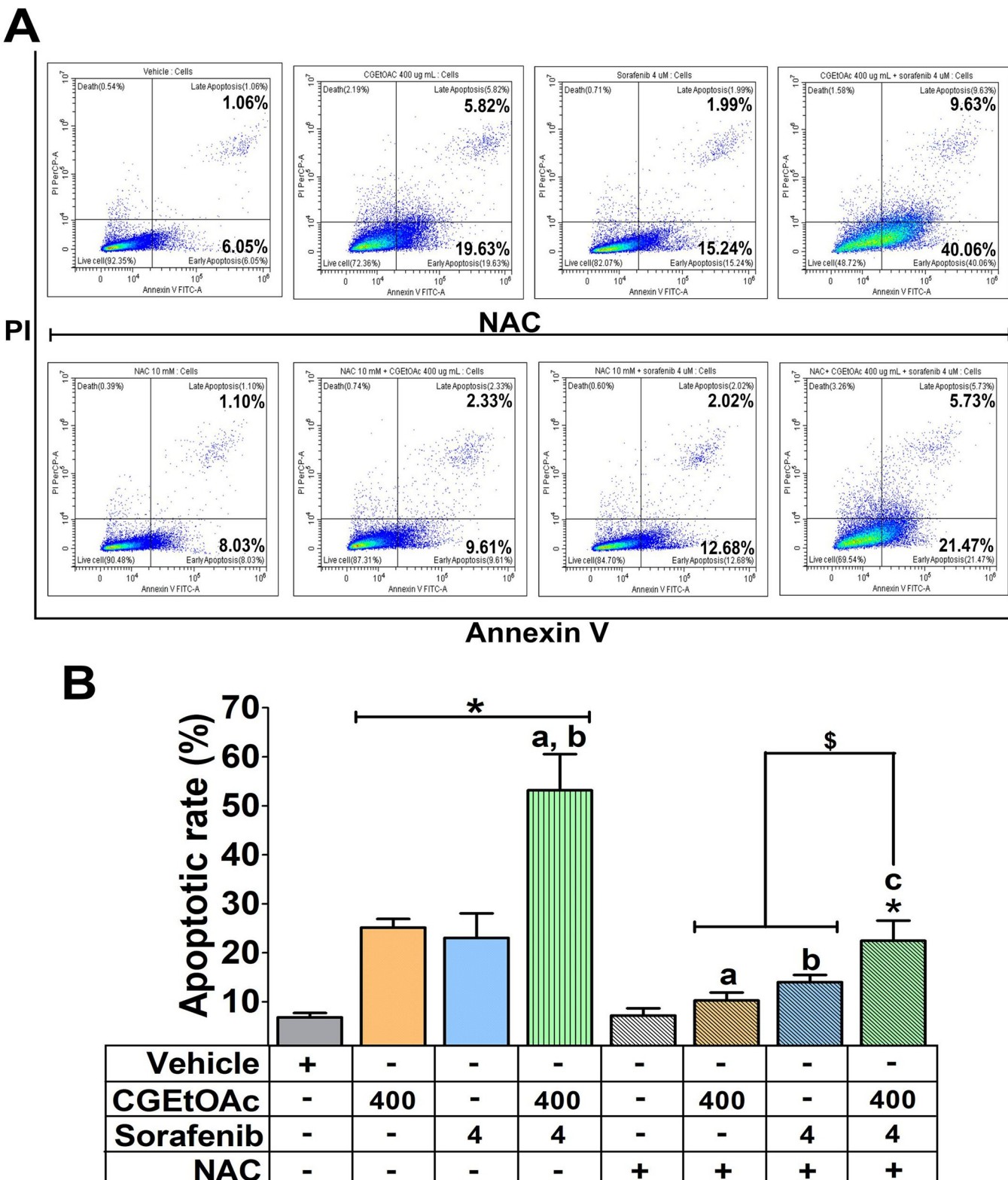

**Fig 6. The apoptosis-inducing effect was associated with ROS generation in HepG2 cells after being exposed to the combination of CGEtOAc and sorafenib for 24 h.** Following N-acetylcysteine (NAC) pre-incubation, cells were exposed to CGEtOAc, sorafenib, and their combination for 24 h. (A) Apoptotic levels were analyzed using annexin V and PI double labeling, followed by flow cytometry. (B) Bar graph depicting the percentage of the total apoptotic rate. Cells treated with 0.8% DMSO represented the vehicle control. The significant differences in data, presented as the mean ± SD from at least

three different experiments, were investigated with a one-way ANOVA using Tukey's HSD test: *; p < 0.05 compared to the vehicle control. [a]; p < 0.05 compared to a single CGEtOAc treatment. [b]; p < 0.05 compared to a single sorafenib treatment. [c]; p < 0.05 compared to a combination treatment of CGEtOAc and sorafenib.

In addition to cardiac glycoside as the major component of the ethyl acetate fraction, other low-level components identified in *C. gigantea*, and other plant extracts have been demonstrated to exhibit anticancer efficacy. Coroglaucigenin from *C. gigantea* stem and leaf extracts exhibited cancer cytotoxicity concomitant with an increase in intracellular ROS in A549

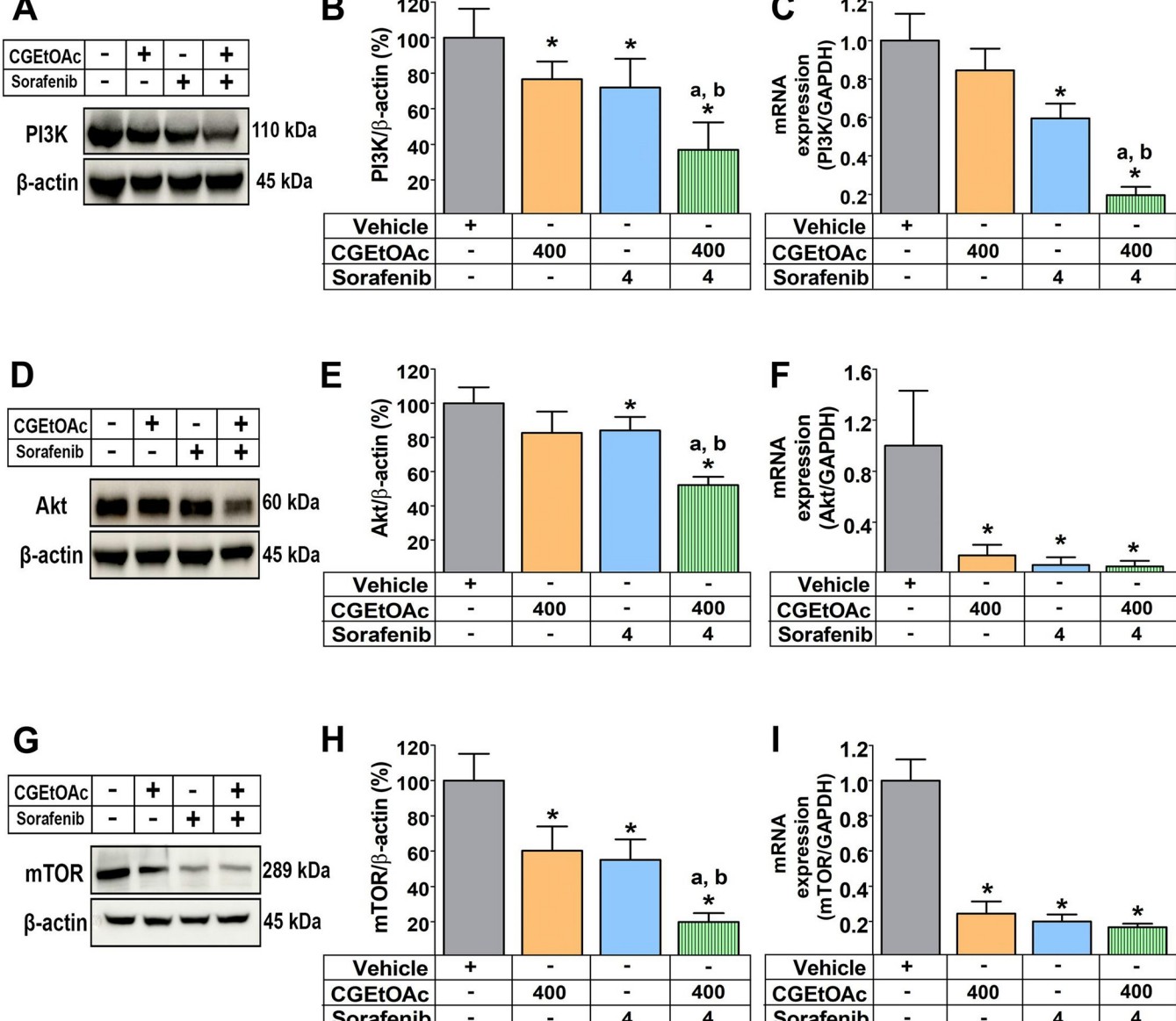

**Fig 7. The combination of CGEtOAc and sorafenib downregulated the expression of the phosphatidylinositol-3-kinase (PI3K)/ protein kinase B (Akt)/ mammalian target of rapamycin (mTOR) signaling pathway after the incubation of HepG2 cells for 24 h.** (A-B) Representative western blot images, bar graph, and quantitative reverse transcription-polymerase chain reaction analysis of the relative expression levels of (A-C) PI3K, (D-F) Akt, and (G-I) mTOR. Cells treated with 0.8% DMSO represented the vehicle control. The significant differences in data, presented as the mean ± SD from at least three different experiments, were investigated with a one-way ANOVA using Tukey's HSD test: *; p < 0.05 compared to the vehicle control. [a]; p < 0.05 compared to a single CGEtOAc treatment. [b]; p < 0.05 compared to a single sorafenib treatment.

human lung cancer cells [46]. Calotroposid A, isolated from the ethyl acetate fraction of *C. gigantea* roots, triggered the extrinsic apoptosis in WiDr colon cancer cells [47]. A cardenolide calotropin from *C. gigantea* downregulated the proliferation of human colorectal cancer cells [27]. Bacaba, a phenolic from the extract of *Oenocarpus bacaba*, promoted apoptosis in breast cancer cells [48]. Pectolinarigenin, a flavonoid isolated from *Cirsium japonioum*, *Eupatorium odoratum*, and *Trollius chinensis*, caused ROS generation-mediated apoptosis in A375 non-pigmented human melanoma cells [19]. Raddeanin A, a triterpenoid, enhanced ROS production and promoted apoptotic cells in non-small cell lung cancer [21].

According to the findings of this study, HepG2 cell exposed to a combination of 400 µg/mL CGEtOAc and 4 µM sorafenib (half of their respective $IC_{50}$) for 24 h demonstrated increased mitochondrial-association with apoptosis. It was discovered that the ROS generation was predominantly responsible for the apoptosis-inducing effects of combined therapy, however, ROS is not the only mechanism participating in the apoptotic action; other mechanisms may work in conjunction to generate this treatment efficiency. Inhibition of PI3K/Akt/mTOR expression also supports the hypothesis that the apoptotic mechanism of CGEtOAc, when combined with sorafenib, may be associated with the downregulation of the upstream PI3K/AKT/mTOR signaling cascade pathway. However, the experiments presented did not provide sufficient evidence of this inhibition in direct association with the apoptotic effects following the treatment. Furthermore, consistent with our findings, there have been reports on the effect of *C. procera* extracts and cardiac glycosides in the inhibition of the expression of the PI3K/AKT/mTOR pathway in correlation to induce apoptosis cancer cells [22, 49–53].

Several studies have established that the apoptotic effect of *C. gigantea* and other plant extracts is primarily related to an upregulation in ROS generation and oxidative damage to cancer cells, for example, a whole plant ethanolic extract treatment in non-small cell lung cancer [15] and the dichloromethane extract fraction of the *C. gigantea* stem bark in HCT116 cells [26]. The apoptotic effect, which primarily involved ROS formation, was also demonstrated in many other extract treatments, for example, in pectolinarigenin (flavonoid) from *Cirsium japonioum*, *Eupatorium odoratum*, and *Trollius chinesis* used against A375 non-pigmented human melanoma cells [19]; a natural flavonoid, apigenin from *Clerodendrum viscosum* leaves used against human breast adenocarcinoma cells [54]; isoorientin from several plants, such as, *Phyllostachys pubescens Patrinia*, and *Drosophyllum lusitanicum* used against HepG2 cells [55]; piperlongumine, an alkaloid from *Piper longum* L. used against MG63 human osteosarcoma cells [24]; berberis, an alkaloid from *Berberis hispanican* root bark used against human laryngeal epidermoid carcinoma Hep-2 [56]; ginsenoside Rk1 used against MCF-7 human breast cancer cell [37]; delicaflavone, a biflavonoid from *Selaginella doederleinii* used against HCT116 and HT29 colorectal cancer cells [57]; 3-deoxysappanchalcone from *Caesalpinia sappan* L. (Leguminosae) used against human esophageal cancer cells KYSE30 and KYSE410 [20]; and raddeanin a, a triterpenoid from *Anemone raddeana Regel* used against non-small cell lung cancer A549 and H1299 cells [21].

Enhanced ROS generation promoted apoptosis by inducing MMP loss in nonpigmented human melanoma cells as a consequence of pectolinarigenin treatment [19]. Similarly, ROS production by 3-deoxysappanchalcone treatment caused MMP loss that resulted in apoptosis in human esophageal cancer cells [20]. In addition, apoptosis mediated by ROS production was found to be involved in activating p53 expression in human breast cancer cells treated with apigenin from *Clerodendrum viscosum* leaves [54] and *Berberis hispanica* alkaloid extracts [56]. Furthermore, ROS levels enhancing apoptosis in non-small cell lung cancer A549 and H1299 cells following treatment with raddeanin A were also reported to increase the phosphorylation of the STAT3 pathway [21]. Triterpenoid pristimerin from Celastraceae and Hippocrateaceae [58], as well as *Musca domestica* pupae [59], upregulated apoptosis via ROS-

induced endoplasmic reticulum (ER) stress in CAL-27 and SCC-15 oral squamous carcinoma and human liver cancer HepG2cells, respectively. Additionally, triterpenoids from *Euphorbia macrostegia* impaired mitochondrial respiration, leading to increased ROS accumulation, ultimately contributing to ER stress and apoptosis in cancer cells [60].

Increased ROS production has been reported as a regulator of PI3K/Akt/mTOR expression, which results in cancer cell apoptosis, examples include kaemperol, a flavonoid found in Chinese herbs belonging to Zingiberaceae [23], piperlongumine isolated from *Piper longum* L. [24], isoorientin treatment [55], and ginsenoside Rk1 [37]. The PI3K/Akt/mTOR (including mTOR complex 1, mTORC1, and mTORC2) signaling pathway regulates apoptosis in many human diseases, including malignancies and proliferative disorders. Phosphatidylinositol 3,4,5-triphosphate is a significant regulator of Akt following the activation of PI3K-activated receptor tyrosine kinases in the treatment of growth stimuli [61]. Activated Akt results in the regulation of downstream substrates to control the cell cycle, growth, proliferation, and energy metabolism, whereas mTOR targets protein kinases that enhance ribosome production and protein synthesis [62]. As a result, inhibiting the PI3K/Akt/mTOR pathway is considered a potentially attractive treatment for cancer cells [61] as reported when the following compounds were isolated: tanshinone I (from *Salvia miltiorrhiza*) [63], lanatoside C [22], isoorientin [55], mahanine [18], piperlongumine [24], and apigetrin [64].

Notably, several studies have reported the apoptotic effect of cardiac glycosides, with this compound identified as one of the major components in *C. gigantea* extracts that exhibit anticancer activity. This effect is achieved through the mechanism of inhibiting $Na^+/K^+$ ATPase, leading to an increase in intracellular $Ca^{2+}$ via the $Na^+/Ca^{2+}$ exchanger [65, 66]. Cardenolides from *C. gigantea* exhibited potent anticancer effects against breast cancers by inhibiting $Na^+/K^+$ ATPase, resulting in an increase in intracellular $Ca^{2+}$ via the $Na^+/Ca^{2+}$ exchanger-dependent manner, ultimately leading to apoptosis [67]. Cardenolide UNBS1450, having lower affinity for the α-2 $Na^+/K^+$ ATPase than other cardenolides, induced downregulation of myeloid cell leukemia-1, a member of the anti-apoptotic protein family downstream of $Na^+/K^+$ ATPase activity, contributing to apoptosis in cancer cells [68]. Various signaling pathways downstream of $Na^+/K^+$ ATPase inhibition has been demonstrated to contribute to the inhibition of cancer cell proliferation. Oleandrin, a cardiac glycoside extracted from the leaves of *Neriaum oleander*, induced accumulation of $Ca^{2+}$, leading to ER stress-mediated cytotoxic effects, enhancing immunogenic cell death in both in vitro and in vivo cancer models [42]. Asclepioside, a natural cardenolide isolated from *Reevesia formosana*, downregulated the oncoprotein c-Myc protein and inhibited the phosphorylation of tumor suppressor protein Rb, resulting in the inhibition of cancer cell cycle progression and blocking cell proliferation. This effect was attributed to an increase of α-tubulin acetylation, which contributed to the inhibition of $Na^+/K^+$ ATPase activity by enhancing the endocytosis of this protein in cancer cells through activation of the mitogen-activated protein kinases (MAPKs) pathway [69].

Additionally, toxicarioside G, a cardenolide isolated from *C. gigantea*, exhibited another mechanism of anticancer activity by inhibiting cancer cell viability and proliferation. It enhanced Yes1 associated transcriptional regulator (YAP) dephosphorylation and nuclear localization, along with downstream target gene expression [70]. Calotropin, a cardiac glycoside derived from *C. gigantea*, demonstrates an inhibitory effect on the regulation of metabolic reprogramming in cancer, specifically targeting aerobic glycolysis (the Warburg effect). This leads to activation of cell cycle arrest and inhibition metastasis in cancer cells, ultimately contributing to activation of apoptosis [71]. Another cardiac glycoside, 3′-epi-12β-hydroxyfroside, isolated from the roots of *C. gigantea*, induced autophagy, leading to enhanced apoptosis. This effect was mediated by downregulation of the heat shock protein 90 (Hsp90)-regulated Akt/ mTOR pathway in lung cancer cells [72]. Lanatoside C, a cardiac glycoside, has been shown to

induce apoptosis in cancer cells by suppressing the Janus kinase (JAK)/signal transducer and activator of transcription (STAT)/cytokine signaling (SOCS) (JAK2/STAT6/SOCS2) pathway [73].

Several cellular pathways have been demonstrated in the anticancer-apoptotic activity of sorafenib. Sorafenib triggers apoptosis by increasing intracellular ROS levels in Bel7402 hepatocellular carcinoma [5] and NCI-H292 human non-small lung cancer cells [8]. Sorafenib suppresses cell invasion and cell proliferation in hepatocellular carcinoma HepG2 and Huh-7 cells by up-regulating p53 and Forkhead box M1 transcription factor expressions, resulting in the inhibition of matrix metalloproteinase 2 and Ki-67 expression [74]. Sorafenib inhibits cyclin D1 expression in NB4 acute promyelocytic leukemia cells [75], SW982, and HS-SY-II synovial sarcoma cells [76]. Sorafenib activates apoptotic cells in human cancer by reducing the levels of the p-Akt, p-mTOR, and p-ERK pathways [5–7, 76, 77]. Furthermore, sorafenib treatment upregulates the expression of phosphorylation-Jun N-terminal kinase and -JUN in human hepatocellular carcinoma Huh-7 cells [78], downregulates the expression of phosphorylation-STAT3 in human oral cancer MC-3 cells [79], and downregulates the expression of the ERK and MEK pathways in HepG2 and PLC/PRF5 cells [80]. Additionally, sorafenib reduces intracellular ATP levels, which may influence the downregulation of the expression of $Na^+/K^+$ ATPase, as has been reported following ouabain administration in cancer cells [81].

Combining sorafenib with anticancer agents promotes cancer cell apoptosis. Artesunate combined with sorafenib had a synergistic effect on HCC cell proliferation by activating ERK and STAT3 signaling pathways [82]. Sorafenib combined with the triterpenoid cucurbitacin from Cucurbitaceae enhanced apoptosis in HepG2 and Huh7 human hepatocellular carcinoma cells by inhibiting STAT3 activity [83]. Sorafenib combined with betulinic acid caused endoplasmic reticulum stress-associated apoptosis in non-small cell lung cancer A549, H358, and A427 cells [84]. Apigenin (4',5,7-trihydroxyflavone) in combination with sorafenib caused extrinsic apoptosis in hepatocellular carcinoma cells [85]. Berberine combined with sorafenib promoted apoptosis in human hepatocellular carcinoma cells [86]. The combination of bufalin and sorafenib enhanced ROS production and MMP-dependent apoptotic response in non-small cell lung cancer NCI-H292 cells [8]. Furthermore, osthole (a coumarin), sorafenib, and the combination of both compounds suppressed PI3K and Raf kinases, resulting in cytotoxic effects in cancer cells [87]. Additionally, sorafenib in combination with capsaicin downregulated the expression levels of Akt, mTOR, and p70S6K in LM3 human hepatocellular carcinoma cells, thereby inducing apoptosis [7].

In addition to the possible mechanism proposed in this study for CGEtOAc, combined with sorafenib-induced apoptosis in cancer cells, which involves increased ROS accumulation and inhibition of PI3K/Akt/mTOR expression, the apoptotic effects resulting from the combination treatment of compounds found in *C. gigantea* extracts and anticancer agents may be attributed to several mechanisms. For instance, the synergistic effect of sorafenib in combination with the $Na^+/K^+$ ATPase inhibitor berbamine, an alkaloid isolated from *Berberis amurensis*, and cardiac glycosides like ouabain, is proposed to be contributed by the potentiation of epidermal growth factor receptor (EGFR)-mediated ERK1/2 and p38MAPK activation by berbamine [88]. Similarly, the combination of digitoxin and sorafenib has been reported to suppress p-ERK, hypoxia inducible factor (HIF)-1α, HIF-2α, and VEGF expression, contributing to apoptosis in cancer cells [89]. Furthermore, the combination of cardenolide derivative, AMANTADIG (3β-[2-(1-amantadine)-1-on-ethylamine]-digitoxigenin) and docetaxel induced apoptosis through the inhibition of surviving protein expression in human androgen-insensitive prostate cancer cells [90].

Thus, the anticancer-apoptotic activity of CGEtOAc and sorafenib was enhanced when administered in combination with HepG2 cells. A reduced dose of both the extract and

sorafenib may be beneficial for future alternative cancer treatments, resulting in fewer adverse effects and a positive outcome. However, the limitation of this study is that it did not evaluate the direct association of the apoptosis-mitochondrial dependent pathway or the regulation of the PI3K/Akt/mTOR pathway in apoptosis following CGEtOAc and sorafenib treatment in cancer cells. Additional research is required to verify the selective mechanism of the cancer therapeutic efficacy of this combination therapy.

## Conclusions

In this study, a combination of CGEtOAc and sorafenib at a dosage half their respective $IC_{50s}$ activated MMP-dependent apoptosis in association with the production of ROS as the major regulator in HepG2 cells. This ROS production was suggested to be related to PI3K/Akt/mTOR pathway suppression. Thus, our findings provide valuable fundamental data for future research on the development of anticancer regimens derived from the *C. gigantea* stem bark extract, the ethyl acetate fraction, and the reduction of the chemotherapy dosage used to treat cancer patients with potentially less severe side effects.

## Supporting information

**S1 Fig. The high-pressure liquid chromatography-electrospray ionisation-mass spectroscopic (HPLC-ESI-MS) chromatogram of *C. gigantea* stem bark extracts (CGEtOAc) in negative mode.**
(PDF)

**S2 Fig. The IC50 curves for (A) CGEtOAc and (B) sorafenib in HepG2 cells, and (C) CGEtOAc in IMR-90 cells for 24 h of incubation.**
(PDF)

**S3 Fig. The combination index (CI) vs. fraction affected (Fa) graph for CGEtOAc in combination with sorafenib after 24 h of incubation.**
(PDF)

**S4 Fig. The migration rate of HepG2 cells treated with CGEtOAc at 400 μg/mL and sorafenib at 4 μM, both singly and in combination, was evaluated using a wound healing assay after 0–72 h of incubation and compared to the vehicle group.** Cells treated with 0.8% DMSO represented the vehicle control. The significant differences in data, presented as the mean ± SD from at least three different experiments, were investigated with a one-way ANOVA using Tukey's HSD test: [a]; $p < 0.05$ compared to 24 h of incubation in the vehicle group, [b]; $p < 0.05$ compared to 48 h of incubation in the vehicle group, and [c]; $p < 0.05$ compared to 72 h of incubation in the vehicle group.
(PDF)

**S1 Raw images. Raw images displaying the gating strategies used in flow cytometry of annexin V and propidium iodide (PI) staining in HepG2 cells after 24 h of incubation, with a combination of 400 μg/mL CGEtOAc and 4 μM sorafenib.**
(PDF)

**S2 Raw images. Raw images of the original uncropped and unadjusted western blot images for HepG2 cells treated with a combination of 400 μg/mL CGEtOAc and 4 μM sorafenib for a 24-h incubation period.**
(PDF)

## Author Contributions

**Conceptualization:** Supawadee Parhira, Piyarat Srisawang.

**Data curation:** Dumrongsak Pekthong, Supawadee Parhira, Piyarat Srisawang.

**Formal analysis:** Pattaraporn Chaisupasakul, Dumrongsak Pekthong, Supawadee Parhira, Piyarat Srisawang.

**Funding acquisition:** Apirath Wangteeraprasert, Supawadee Parhira, Piyarat Srisawang.

**Investigation:** Pattaraporn Chaisupasakul, Dumrongsak Pekthong, Worasak Kaewkong, Supawadee Parhira, Piyarat Srisawang.

**Methodology:** Pattaraporn Chaisupasakul, Dumrongsak Pekthong, Worasak Kaewkong, Naphat Kaewpaeng, Supawadee Parhira, Piyarat Srisawang.

**Project administration:** Supawadee Parhira, Piyarat Srisawang.

**Resources:** Piyarat Srisawang.

**Software:** Pattaraporn Chaisupasakul.

**Validation:** Pattaraporn Chaisupasakul, Supawadee Parhira, Piyarat Srisawang.

**Visualization:** Pattaraporn Chaisupasakul, Supawadee Parhira, Piyarat Srisawang.

**Writing – original draft:** Pattaraporn Chaisupasakul, Supawadee Parhira, Piyarat Srisawang.

**Writing – review & editing:** Pattaraporn Chaisupasakul, Dumrongsak Pekthong, Apirath Wangteeraprasert, Worasak Kaewkong, Julintorn Somran, Naphat Kaewpaeng, Supawadee Parhira, Piyarat Srisawang.

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
