## [Decision Letter · Decision Letter 0]

12 Sep 2023

PONE-D-23-18216

Enhanced apoptosis of sorafenib in HepG2 cells by combining with the ethyl acetate fraction from the Calotropis gigantea stem bark extract

PLOS ONE

Dear Dr. Srisawang,

Thank you for submitting your manuscript to PLOS ONE. After careful consideration, we feel that it has merit but does not fully meet PLOS ONE’s publication criteria as it currently stands. Therefore, we invite you to submit a revised version of the manuscript that addresses the points raised during the review process.

We look forward to receiving your revised manuscript.

Kind regards,

Nafees Ahemad

Academic Editor

PLOS ONE

Journal Requirements:

   "SP and PS received grant supported from National Science Research and Innovation Fund (NSRF) of Thailand [Grant NO. R2564B007]. 

AW received grant supported from National Science Research and Innovation Fund (NSRF) of Thailand [Grant NO. R2564B033]. 

SP received grant supported from Agricultural Research Development Agency (Public Organization) [Grant NO. CRP6505030030]. 

PC received grant supported from Center of Excellence for Innovation in Chemistry (PERCH-CIC) [Grant NO. NUPMEM02/63] and Faculty of Medical Science, Naresuan University, Phitsanulok, Thailand [Grant NO. 63064382]."

Additional Editor Comments:

Please refer to reviewers comments.

Please see the comments by reviewers. The manuscript may be accpeted subject to substantial revision is done and resubmitted..

Reviewers' comments:

Reviewer's Responses to Questions

**Comments to the Author**

1. Is the manuscript technically sound, and do the data support the conclusions?

Reviewer #1: Yes

Reviewer #2: Partly

2. Has the statistical analysis been performed appropriately and rigorously? 

Reviewer #1: I Don't Know

Reviewer #2: Yes

3. Have the authors made all data underlying the findings in their manuscript fully available?

Reviewer #1: Yes

Reviewer #2: Yes

4. Is the manuscript presented in an intelligible fashion and written in standard English?

Reviewer #1: Yes

Reviewer #2: No

5. Review Comments to the Author

Reviewer #1: In this study, authors examined the effect of combination treatment of Sorafenib and CGEtOAc on HepG2 cells, aiming that combination treatment maximizes the effect of killing cancer cells and minimizes side-effects.

Here are several points need to be clarified:

[1] In Fig.1D, * (asterisk) applies to all bars starting 200 to 2,000µg/ml; therefore it appears to be statistically significant. However, in the text, 1,000~2,000 are not significant. Which one is accurate?

[2] Fig.1E: Does CGE treatment more toxic in normal fibroblasts than Sorafenib?

[3] The intensity of Hoechst 33342 staining seems to be low in vehicle control and NAC-treated samples (Fig.3A). Why?

[4] Discussion is very long (7 pages). It would be nice to state briefly.

Reviewer #2: This manuscript by Chaisupasakul et al. investigates the potential of the ethyl acetate fraction of the stem bark extract of Calotropis gigantea for use in combination therapy with sorafenib against HepG2 cells. They showed this by testing the cytotoxicity caused by each agent singly, and in combination. They have also shown that the combination of the extract and sorafenib induces early apoptosis. Possible related mechanisms have been explored such as ROS production and PI3K/Akt/mTOR pathway inhibition. This is a good starting point; however, the study presented still lacks some key experiments that would solidify the conclusions proposed. Also, further compound purification should be explored to support the combinatorial experiments. Major rewriting of the paper is also recommended so that important points can be clear to the reader as the statements in its current form have grammatical errors ad are confusing. Hence, the manuscript in its current form is not recommended for publication yet. Further elaboration of the review, as well as suggestions to improve the manuscript is written below.

1) Change short title to C. gigantea stem bark ethyl acetate fraction combined “with” sorafenib induces apoptosis in HepG2 cells.

2) Shorten keyword “The ethyl acetate fraction of Calotropis gigantea stem bark extract”.

3) The statement “The leaf and root ethanolic extract of C. gigantea enhanced cytotoxicity in T47D breast cancer cells” should be replaced with “The leaf and root ethanolic extract of C. gigantea exhibited cytotoxicity in T47D breast cancer cells”.

4) Change “curative properties on cancer” or “cancer curative” to “anti-cancer properties”.

5) Room temperature stated in the methods is not standard room temperature.

6) Define EA.

7) There are a lot of missing statements for reagent sources or suppliers and concentrations used for the experiments in the methodology section. Please take a closer look and fix.

8) “under a humidified 5% CO2 at 37°C incubator” should be “humidified incubator at 5% CO2, 37°C”.

9) Include seeding densities for all cell lines used for all of the experiments.

10) State what reservoir the cells were cultured for testing the test compounds and extracts.

11) Was there no solubilization step of the resulting formazan pellet once MTT is added?

12) Specify how the wounds were made for the cell migration assay.

13) Indicate model of flow cytometer used.

14) Indicate concentrations of the stains and antibodies used.

15) Complete statement header for qRT-PCR. What are you running PCR with?

16) Show the list of mRNA templates in Supplementary Information.

17) Indicate concentration of primers used.

18) Table 1 title should be a complete statement.

19) The note in Table 1 can be removed since it should be stated in the methods section.

20) Indicate incubation time, antibody concentration, and what buffer was used for Western blots.

21) There should at least be an MS profile, HPLC trace, or proof of further purification efforts in a Figure or Supplementary Information since the components of CGEtOAc may contain known compounds that have already been reported to synergize with sorafenib.

22) The curve and statistical model used to compute for the IC50 should be shown in a Figure, as well as methods.

23) The IC50 of the extracts seemed a bit high for exhibiting cytotoxic effects. This may still show potential but is mostly not considered as a candidate in the long run. This may be solved if the causative compound is isolated.

24) Figure 1 C would be better visualized if it were presented as a heat map to show a checkerboard analysis.

25) The fractional inhibitory concentration should be computed to determine if the combinatorial effect is either synergistic, potentiation, or just an additive effect.

26) According to Figure 1 E, CGEtOAc is still cytotoxic in normal cells tested. Hence, the IC50 of CGEtOAc should be computed and a therapeutic index determined.

27) Migration studies should be placed as a separate figure.

28) The results shown in Figure 1E and 1G did not show that the gap in the vehicle and CGEtOAc-treated cells were different and hence conclusion from this is not supported. The gap made from the wound healing assay for vehicle alone is still comparable with the baseline, hence, this assay should still be optimized before using any test compounds and extracts.

29) For all flow cytometry experiments, gating paradigms prior to the final image should be shown in the Supplementary Information.

30) Show how many events were collected for all flow cytometry experiments.

31) The statement “the combination of CGEtOAc and sorafenib promoted apoptosis in HepG2 cells depending on MMP damage” is not fully supported by the results presented. All the results have shown is that the combination of CGEtOAc and sorafenib causes early apoptosis and MMP disruption but it doesn’t necessarily mean that they are directly related to each other or the events are sequential. A temporal aspect to the experiment could be introduced to determine which events come first.

32) JC-1 and Hoechst 33342 imaging should be a separate Figure.

33) The header “The combination of CGEtOAc and sorafenib triggered HepG2 apoptosis through increasing cellular ROS levels” should be changed as this experiment only proves that early apoptosis caused by CGEtOAc and sorafenib combination is reduced when cellular ROS is blocked but it does not fully support that increased ROS happens before early apoptosis.

34) In Figure 3B, despite being statistically significant, the ROS levels are already high even if it is just in the single treatment of sorafenib and CGEtOAc, hence, the combination of the two triggering an increase of ROS levels may not be necessarily true and this is just an additive effect rather than synergy.

35) Figure legends stating that these are histograms should be changed as all data shown are in bar graphs.

36) The statement “Additionally, when ROS generation was inhibited, the apoptotic effect of CGEtOAc, sorafenib, and their combination was considerably reduced to baseline levels…” should be changed as they are reduced but not to baseline levels.

37) Indicate what $ mean in Figure 4B.

38) Indicate concentration of NAC used.

39) There is no direct evidence that the downregulation of PI3K/Akt/mTOR proteins and mRNA causes early apoptosis, hence, the statement that CGEtOAc and sorafenib combination causes early apoptosis thru PI3K/Akt/mTOR inhibition should be reconstructed and conclusions be changed.

40) Explain clearly why would the cytotoxicity of each fraction from C. gigantea ST is not entirely influenced by a single component present in each fraction.

41) If it is hypothesized that a cardiac glycoside would be a key component in the EA fraction of C. gigantea ST extract, this should be isolated and further tested in the battery of assays.

6. PLOS authors have the option to publish the peer review history of their article (what does this mean?). If published, this will include your full peer review and any attached files.

Reviewer #1: No

Reviewer #2: No

---

## [Author Response · Author response to Decision Letter 0]

27 Oct 2023

Journal Requirements:

Response: 

Thank you for these suggestions.

We stated, "Role of Funder statement:

“The funders had no role in study design, data collection and analysis, decision to publish, or preparation of the manuscript."

This statement was also included in the cover letter.

Response: 

Thank you for these valuable suggestions.

We stated the Data Availability statement as follows:

“Data availability will be provided, including repository information for the data, when the paper is accepted for publication, along with the relevant accession numbers or DOIs necessary for access.”

This statement was also included in the cover letter.

Response: 

Thank you for these suggestions.

In the revised manuscript submission, we included the original, uncropped, and unadjusted images that underlie all protein blot and other raw image results reported in the Supporting Information files. The revised manuscript has 6 figure files in the Supporting Information, as addressed in the following. This ensures that these figures fully adhere to the journal's policy and requirements for blot reporting. We also mentioned in the cover letter that we included the original, uncropped, and unadjusted images for all blot and raw image results, including 6 file images in the Supporting Information files.

Supporting information

S1 Fig: The high-pressure liquid chromatography-electrospray ionisation-mass spectroscopic (HPLC-ESI-MS) chromatogram of CGEtOAc in negative mode.

S2 Fig: The IC50 curve of (A) CGEtOAc and (B) sorafenib in HepG2 cells, and (C) CGEtOAc in IMR-90 cells after 24 h of incubation.

S3 Fig: The combination index (CI) vs. fraction affected (Fa) graph of CGEtOAc in combination with sorafenib after 24 h of incubation.

S4 Fig: The migration rate of CGEtOAc at 400 µg/mL in combination with 4 µM sorafenib in HepG2 cells after 24 h of incubation.

S5 Raw images displaying the gating strategies used in flow cytometry of annexin V and propidium iodide (PI) staining in HepG2 cells after 24 h of incubation with a combination of 400 µg/mL CGEtOAc and 4 µM sorafenib.

S6 Raw images of the original uncropped and unadjusted western blot images for HepG2 cells treated with a combination of 400 µg/mL CGEtOAc and 4 µM sorafenib for a 24-h incubation period.

Response: 

Thank you for these valuable suggestions.

In the Methods section of the revised manuscript, we stated and provided the full name of the ethics committee that approved the human cell culture research as follows:

“The human cell culture research was approved by the Naresuan University Institutional Review Board (NU-IRB) in Panel 1: Health Sciences, with the approval number P1-0166/2565."

 

Reviewers' comments:

Dear Reviewers,

Thank you very much for your valuable suggestions and comments on our manuscript. These comments have been immensely helpful in improving and revising our paper. We have carefully studied your comments and made corrections in line with your suggestions. The revised portions are marked in blue in the paper. The main corrections in the paper and our responses to the reviewers’ comments and remarks are as follows:

1. Is the manuscript technically sound, and do the data support the conclusions?

Reviewer #1: Yes

Reviewer #2: Partly

Response:

We appreciate the positive comments from the reviewers.

2. Has the statistical analysis been performed appropriately and rigorously?

Reviewer #1: I Don't Know

Reviewer #2: Yes

Response: 

We are grateful for the favorable remarks provided by the reviewers.

3. Have the authors made all data underlying the findings in their manuscript fully available?

Reviewer #1: Yes

Reviewer #2: Yes

Response: 

We are grateful for the favorable remarks provided by the reviewers.

4. Is the manuscript presented in an intelligible fashion and written in standard English?

Reviewer #1: Yes

Reviewer #2: No

Response:

We express our gratitude for the positive comments and valuable suggestions provided by the reviewers.

5. Review Comments to the Author

 

Reviewer #1:

In this study, authors examined the effect of combination treatment of Sorafenib and CGEtOAc on HepG2 cells, aiming that combination treatment maximizes the effect of killing cancer cells and minimizes side-effects.

Here are several points need to be clarified:

Response:

We appreciate the reviewer's comprehensive and insightful analysis of our paper, as well as constructive and necessary suggestions.

This manuscript represents an ongoing project aimed at conducting purification experiments and investigating potential biomarker compounds in the promising extract from C. gigantea stem barks. We propose the potential of the EtOAc fraction of C. gigantea stem barks and publish the possibility of using natural products in combination with the chemotherapeutic agent sorafenib for further research into future anticancer agents. It requires additional experimental phases and substantial support to achieve our objective. We hope that our manuscript will meet the journal's standards and provide valuable knowledge worldwide. We sincerely regret any grammatical errors and typos; throughout the manuscript, they have been carefully corrected.

The revised manuscript has been thoroughly rechecked, the English has been meticulously reviewed, and it now meets the required standards for publication with the support of Editage's editorial services (www.editage.com). An editing certification is also attached.

[1] In Fig.1D, * (asterisk) applies to all bars starting 200 to 2,000µg/ml; therefore it appears to be statistically significant. However, in the text, 1,000~2,000 are not significant. Which one is accurate?

Response:

Thank you for your question, which provides valuable feedback and prompts us to provide a clearer explanation of this issue in the revised manuscript.

Figure 1D demonstrates that the toxicity of CGEtOAc to IMR-90 cells at concentrations ranging from 200 to 2,000 µg/mL is significant compared to the vehicle. However, these concentrations are less cytotoxic to IMR-90 cells than what was observed in HepG2 cells. Notably, CGEtOAc at 1,000, 1,500, and 2,000 µg/mL did not show a significant difference from the effect at 800 µg/mL. This finding suggests that from 200 µg/mL to the highest dose of CGEtOAc at 2,000 µg/mL, it exhibits lower cytotoxicity to normal IMR-90 cells while causing a dose-dependent increase in cytotoxicity to HepG2 cells.

[2] Fig.1E: Does CGE treatment more toxic in normal fibroblasts than Sorafenib?

Response:

Thank you for your question.

For a 24-hour incubation period, the IC50 of CGEtOAc on IMR-90 cells was greater than 5,000 µg/mL. We did not evaluate the IC50 of sorafenib on IMR-90 cells. In contrast to IMR-90 cells, the IC50 of CGEtOAc and sorafenib on HepG2 cells were 771.00 ± 69.74 µg/mL and 8.64 ± 0.32 µM, respectively, as shown in Figure 1. We then used half-IC50 doses of CGEtOAc at 400 µg/mL and sorafenib at 4 µM, as well as a combination of both to treat IMR-90 cells.

As observed, the toxicity of CGEtOAc at 200, 400, and 800 µg/mL to IMR-90 cells was significant compared to the vehicle, although it was less pronounced than that observed in HepG2 cells, as shown in Figure 1E of the revised manuscript. Additionally, the combination of CGEtOAc at 400 µg/mL and sorafenib at 4 µM had a significant cytotoxic effect on IMR-90 cells compared to the vehicle control, but it was less pronounced than the effect in HepG2 cells (as seen in Figure 1F of the revised manuscript). This combination also caused more cytotoxicity in HepG2 cells, resulting in approximately 60% inhibition of cell viability, as shown in Figure 1C of the revised manuscript.

In summary, when using the same doses of CGEtOAc and CGEtOAc in combination with sorafenib, it exhibited less cytotoxicity to IMR-90 cells, while causing more cytotoxicity to cancer cells.

[3] The intensity of Hoechst 33342 staining seems to be low in vehicle control and NAC-treated samples (Fig.3A). Why? CGEtOAc 400 µg/mL and sorafenib 4 µM

Response:

We are grateful for the favorable remarks provided by the reviewers.

Hoechst 33342 is used to assess cell death by apoptosis, characterized by DNA fragmentation or nuclear condensation. The vehicle or the vehicle with NAC treatment showed fewer apoptotic specific hallmarks, suggesting reduced ROS-induced apoptosis, as indicated in this manuscript. Consequently, the reduced staining of cells with Hoechst 33342 resulted in fewer apoptotic cells.

Example references with their DOIs:

https://doi.org/10.2147/DDDT.S189969

https://doi.org/10.1111/jmi.12133

https://doi.org/10.1002/jbt.22616

https://doi.org/10.1093/jpp/rgac037

[4] Discussion is very long (7 pages). It would be nice to state briefly.

Response:

We appreciate the reviewer's insightful comments and suggestions for improving the readability of the manuscript.

The revised manuscript has been edited, and the discussion section has been modified to be more concise.

We are genuinely appreciative of the Editor and Reviewers’ decision to allow us the opportunity to revise this manuscript to fully meet PLOS ONE's publication criteria. We are thankful for the time, effort, and assistance you have provided in enhancing the quality of this work. We sincerely hope that our revisions have elevated the manuscript's quality, and we are confident that the reviewers will find it suitable for publication.

Sincerely yours,

On behalf of all co-authors,

Corresponding authors

 

Reviewer #2:

This manuscript by Chaisupasakul et al. investigates the potential of the ethyl acetate fraction of the stem bark extract of Calotropis gigantea for use in combination therapy with sorafenib against HepG2 cells. They showed this by testing the cytotoxicity caused by each agent singly, and in combination. They have also shown that the combination of the extract and sorafenib induces early apoptosis. Possible related mechanisms have been explored such as ROS production and PI3K/Akt/mTOR pathway inhibition. This is a good starting point; however, the study presented still lacks some key experiments that would solidify the conclusions proposed. Also, further compound purification should be explored to support the combinatorial experiments. Major rewriting of the paper is also recommended so that important points can be clear to the reader as the statements in its current form have grammatical errors ad are confusing. Hence, the manuscript in its current form is not recommended for publication yet. Further elaboration of the review, as well as suggestions to improve the manuscript is written below.

Response:

We appreciate the reviewer's comprehensive and insightful analysis of our paper, as well as constructive and necessary suggestions.

This manuscript represents an ongoing project aimed at conducting purification experiments and investigating potential biomarker compounds in the promising extract from C. gigantea stem barks. We propose the potential of the EtOAc fraction of C. gigantea stem barks and publish the possibility of using natural products in combination with the chemotherapeutic agent sorafenib for further research into future anticancer agents. It requires additional experimental phases and substantial support to achieve our objective. We hope that our manuscript will meet the journal's standards and provide valuable knowledge worldwide. We sincerely regret any grammatical errors and typos; throughout the manuscript, they have been carefully corrected.

The revised manuscript has been thoroughly rechecked, the English has been meticulously reviewed, and it now meets the required standards for publication with the support of Editage's editorial services (www.editage.com). An editing certification is also attached.

 

1) Change short title to C. gigantea stem bark ethyl acetate fraction combined “with” sorafenib induces apoptosis in HepG2 cells.

Response:

We appreciate the reviewer's thorough and thoughtful analysis of our paper, as well as the constructive comments.

The title of the revised manuscript has been changed to "Combination of Ethyl acetate fraction from Calotropis gigantea stem bark and sorafenib induces apoptosis in HepG2 cells." We hope the reviewers will agree.

2) Shorten keyword “The ethyl acetate fraction of Calotropis gigantea stem bark extract”.

Response:

Thank you for your suggestions.

The keywords in the revised manuscript include “The ethyl acetate fraction, The Calotropis gigantea stem bark”.

3) The statement “The leaf and root ethanolic extract of C. gigantea enhanced cytotoxicity in T47D breast cancer cells” should be replaced with “The leaf and root ethanolic extract of C. gigantea exhibited cytotoxicity in T47D breast cancer cells”.

Response:

Thank you for your valuable suggestions.

We have edited this statement as suggested.

4) Change “curative properties on cancer” or “cancer curative” to “anti-cancer properties”.

Response:

Thank you for your suggestions.

The revised manuscript has been edited as suggested.

5) Room temperature stated in the methods is not standard room temperature.

Response:

Thank you for your constructive comments.

The revised manuscript has been edited as suggested. The statement is as follows: (Page ...): In brief, the C. gigantea ST was harvested, dried at a temperature of 35 ± 7 ºC, and ground into a powder.

6) Define EA.

Response:

Thank you for your comments. Please accept our apologies for this misspelling.

The ethyl acetate fraction of the Calotropis gigantea stem bark extract has been abbreviated as “CGEtOAc” throughout the revised manuscript.

7) There are a lot of missing statements for reagent sources or suppliers and concentrations used for the experiments in the methodology section. Please take a closer look and fix.

Response:

Thank you for your comments. We apologize for this error.

The revised manuscript has been edited thoroughly.

8) “under a humidified 5% CO2 at 37°C incubator” should be “humidified incubator at 5% CO2, 37°C”.

Response:

Thank you for your constructive suggestions.

The revised manuscript has been thoroughly edited as suggested.

9) Include seeding densities for all cell lines used for all of the experiments.

Response:

Thank you for your constructive suggestions.

The revised manuscript has been thoroughly edited as suggested.

10) State what reservoir the cells were cultured for testing the test compounds and extracts.

Response:

Thank you for your valuable suggestions.

The revised manuscript has been thoroughly edited, and we have addressed this issue as suggested in the Cell Culture Method section.

"Cells were cultured in T-25 culture flasks in complete media and sub-cultured after reaching 80-90% confluency every 4 days of the incubation cycle. Cell numbers were monitored during each subculture to measure the consistency of cell growth rates before proceeding with the cell treatment plating procedure. The human cell culture research was approved by the Naresuan University Institutional Review Board (NU-IRB) in Panel 1: Health Sciences, with the approval number P1-0166/2565."

11) Was there no solubilization step of the resulting formazan pellet once MTT is added?

Response:

We very much appreciate the valuable suggestions from the reviewer.

The revised manuscript has been thoroughly edited, and we have addressed this issue as suggested in the MTT assay method.

“After incubating HepG2 cells at a density of 1.5×104 cells per well/150 µL in a 96-well plate for 24 hours and exposing them to the extracts for an additional 24 hours, we applied 3-(4,5-Dimethylthiazol-2-yl)-2,5-Diphenyltetrazolium Bromide (MTT) at a concentration of 2 mg/ml to assess viable cells based on mitochondrial reductase activity. We then measured absorbance at 595 nm using DMSO to dissolve the formazan crystals produced by MTT, with a microplate reader (SpectraMax ABS, Molecular Devices, USA). The IC50 values were calculated using Graph Prism Software version 9.”

12) Specify how the wounds were made for the cell migration assay.

Response:

Thank you for your constructive suggestions.

The revised manuscript has been thoroughly edited, and we have addressed this issue as suggested in the “Analysis of Cell Migration Activity by Wound Healing Method” section.

“After incubating HepG2 cells at a density of 4×10^5 cells per well/1 mL in a 12-well plate and allowing them to reach 80-90% confluency within 48 hours, we induced the formation of physical wounds in the cell monolayer by scratching the attached cells in a continuous straight line at the center of each well using a 200 µL sterile tip. The scratch wounds were consistently of the same size in each well to minimize variations between conditions (https://doi.org/10.3389/fcell.2019.00107 , https://doi.org/10.3791%2F51046 , https://doi.org/10.3389/fcell.2021.640972 , https://doi.org/10.1186/s40360-018-0284-4 ).

We removed debris and non-attached cells before subsequently exposing the cells to extracts, sorafenib, and their combination for 0-72 hours.

The gap width of the wound was measured and recorded in three regions, with triplicate measurements of each region performed to obtain the mean value. This was done using an inverted microscope (IX71, Olympus, Japan) to measure the distance between the closest points on both sides of the wound. The results were analyzed as the percentage of the scratch gap using cellSens Standard [Ver.2.3] software. The scratch widths of the control and treatment wells were measured and normalized according to their respective values at 0 h. The migrated distance was defined by the following formula:

Migrated rate (%) = (the width of the initial wound - the width of the remaining wound) / the width of the initial wound × 100 (https://doi.org/10.1016/j.phymed.2019.153112 , https://doi.org/10.1186/s12885-022-09684-0).” These results are presented in the Supporting Information (S4 Fig).

13) Indicate model of flow cytometer used.

Response:

Thank you for your suggestions.

The revised manuscript has been thoroughly edited, and we have incorporated this issue as suggested in the “Determination of Apoptosis by Flow Cytometric Analysis” method.

14) Indicate concentrations of the stains and antibodies used.

Response:

Thank you for your suggestions.

The revised manuscript has been thoroughly edited, and we have addressed this issue as suggested.

15) Complete statement header for qRT-PCR. What are you running PCR with?

Response:

Thank you for your suggestions. 

The revised manuscript has been thoroughly edited, and we have updated the header to read: “Real-time quantitative reverse transcription polymerase chain reaction (RT-qPCR) for PI3K/Akt/mTOR gene expression.”

16) Show the list of mRNA templates in Supplementary Information.

Response:

Thank you for your suggestions.

We conducted a real-time RT-PCR assay, consisting of three essential steps:

(i) cellular RNA extraction for the conversion of RNA into cDNA templates by reverse transcriptase,

(ii) the amplification of the cDNA, and

(iii) the real-time detection and quantification of amplification products.

Specific gene primers for PI3K, Akt, and mTOR are detailed in the table provided in the methods section.

The groups of cells studied for gene expression were treated with CGEtOAc at 400 µg/mL, 4 µM sorafenib, and a combination of both. RNA was extracted from these groups to convert it into cDNA templates, which were then compared to the vehicle control group.

17) Indicate concentration of primers used.

Response:

Thank you for your suggestions.

The revised manuscript has been thoroughly edited to include the primer concentration (10 pg/µL) in the method section.

18) Table 1 title should be a complete statement.

Response:

Thank you for your suggestions.

The revised manuscript has been thoroughly edited, and the title of Table 1 has been changed to “The Primer Sequences Used in RT-qPCR”.

19) The note in Table 1 can be removed since it should be stated in the methods section.

Response:

Thank you for your feedback.

The revised manuscript has been meticulously edited in accordance with your suggestions.

20) Indicate incubation time, antibody concentration, and what buffer was used for Western blots.

Response:

Thank you for your comments.

The revised manuscript has been thoroughly edited and now includes the suggested issue.

21) There should at least be an MS profile, HPLC trace, or proof of further purification efforts in a Figure or Supplementary Information since the components of CGEtOAc may contain known compounds that have already been reported to synergize with sorafenib.

Response:

We sincerely appreciate your valuable suggestions.

The HPLC chromatogram of CGEtOAc, along with negative mode mass spectra and m/z values of major ions, has been included in the Supporting Information (S1 Fig). However, due to the lack of other standards to confirm the corresponding peaks and ions, we have reported m/z values and possible ions corresponding to compounds found in Calotropis species. These compounds include the formic acid adduct ions [HCOO]- of Calactinic acid methyl ester (a singly-linked cardenolide) at m/z 607.27 and calactin or calotropin (an isomer, a doubly-linked cardenolide) at m/z 577.26.

Synergistic effects of certain cardiac glycosides, such as digoxin and ouabain (singly-linked cardenolides), have been previously reported.

S1 Fig: The high-pressure liquid chromatography-electrospray ionisation-mass spectroscopic (HPLC-ESI-MS) chromatogram of CGEtOAc in negative mode.

The HPLC-ESI-MS chromatogram of CGEtOAc was obtained from an Agilent 6540 UHD LCMS instrument. CGEtOAc was dissolved in methanol (MS grade, 5 mg/mL, 10 μL) then injected into a stationary phase using a Phenomenex Luna® 3 μm C18 column (150 mm × 4.6 mm i). The temperatures of autosampler and column were 4 and 25 °C, respectively. A mobile phase was a gradient solution of 0.1% formic acid-water solution and 0.1% formic acid-acetonitrile solution (5–95% acetonitrile, 20 min, 0.8 mL/min flow rate). The electrospray ionization mass spectrometry in negative mode (m/z range 200–800) setup with a 30 psi nebulizer pressure (N2), a 10 L/min drying gas flow rate, and a 350 °C temperature was performed. The retention time in the base peak chromatogram (BPC) and mass spectra of possible m/z of CGEtOAc were illustrated.

This manuscript is part of an ongoing project focused on purifying compounds and exploring potential biomarkers in the promising extract from C. gigantea stem barks. Additional experimental phases and substantial support are needed to accomplish our objectives. Currently, we have isolated calactinic acid methyl ester and plan to conduct further experiments in the future.

References: 

1. Xiao Y, Yan W, Guo L, Meng C, Li B, Neves H, Chen PC, Li L, Huang Y, Kwok HF, Lin Y. Digitoxin synergizes with sorafenib to inhibit hepatocelluar carcinoma cell growth without inhibiting cell migration. Mol Med Rep. 2017 Feb;15(2):941-947. doi: 10.3892/mmr.2016.6096. Epub 2016 Dec 30. PMID: 28035421.

2. Yang, S., Yang, S., Zhang, H., Hua, H., Kong, Q., Wang, J., & Jiang, Y. (2021). Targeting Na+/K+-ATPase by berbamine and ouabain synergizes with sorafenib to inhibit hepatocellular carcinoma. British Journal of Pharmacology, 178(21), 4389–4407. https://doi.org/10.1111/bph.15616]

22) The curve and statistical model used to compute for the IC50 should be shown in a Figure, as well as methods.

Response:

We appreciate your valuable suggestions.

The revised manuscript has been updated to include the Supporting Information, which features the IC50 graph of CGEtOAc and sorafenib in HepG2 cells, as well as the IC50 of CGEtOAc in IMR-90 cells (S2 Fig).

23) The IC50 of the extracts seemed a bit high for exhibiting cytotoxic effects. This may still show potential but is mostly not considered as a candidate in the long run. This may be solved if the causative compound is isolated.

Response:

We thank the reviewer for careful and thorough assessment of our manuscript and for providing detailed, valuable, and constructive comments that have enhanced the quality of our work.

We agree with the reviewer's comment. In the first experiment, we screened CGEtOAc for its potential anticancer activity. After discovering its cytotoxicity, we conducted several purification experiments and obtained some pure compounds. These compounds are currently being enriched and their chemical structures elucidated for further investigation.

The fractions of C. gigantea obtained from solvents with varying polarity properties resulted in each fraction containing different phytochemicals. These C. gigantea fractions exhibited a variety of phytochemicals, such as phenolics, triterpenoids, cardiac glycosides, and flavonoids, each with distinct biological activities due to their chemical structures, physicochemical properties, and affinities to target proteins. Consequently, the biological activities, including cytotoxicity, of each fraction were attributed to a combination of chemicals, as previously mentioned and discussed in our report (Winitchaikul T et al., 2021, Sawong S et al., 2022).

After conducting the study on the cytotoxic effects of the extracts and fractions on cancer cells, it was observed that the IC50 values were higher than those of the isolated compounds. Several factors influence the response in various cancer cell models, including the incubation time, the plant parts from which the extract fractions were obtained, and variations in cell models.

Mutiah R. et al. (2018) reported varying IC50 values for the fractionated ethanol extract of C. gigantea roots: ethyl acetate fraction (IC50 0.063 μg/ml), dichloromethane fraction (IC50 0.367 μg/ml), butanol fraction (IC50 12.18 g/ml), and water fraction (IC50 8,493 μg/ml) in cancer cells (https://doi.org/10.22034/APJCP.2018.19.6.1457 ). Our study is consistent with these findings, as a 24-hour incubation of the ethanol extract from the whole plant of C. gigantea demonstrated less than a 20% cytotoxic effect on cancer cells at 15 μg/ml, as reported by Lee J. et al. (2019) (https://doi.org/10.1186/s12906-019-2561-1 ). The cytotoxic activity of the C. gigantea latex showed IC50 values ranging from 400-150 µg after a 24-hour incubation against A549 cancer cells (https://doi.org/10.1016/j.biopha.2016.12.133 ).

Our previous reports demonstrated that in HepG2, HCT116, and HT-29 cells, the IC50 of the extract fractions from the stem bark of C. gigantea was the lowest in the DCM (dichloromethane) fraction, followed by the EtOAc (ethyl acetate), EtOH (ethanol), and water fractions (https://doi.org/10.1038/s41598-022-16321-0, https://doi.org/10.1371/journal.pone.0254392 ).

It has been reported that compounds isolated from the leaves of C. gigantea exhibit varying cytotoxic effects against cancer cells. These effects are influenced by factors such as the polarity of solvents, chemical structures, and the physicochemical properties of the isolated compounds. The presence of an aromatic methoxy group in pinoresinol results in decreased cytotoxicity, with an IC50 greater than 100 µM. Similarly, the presence of a (6-O-vanilloyl)-β-D-glucopyranosyl group decreases the cytotoxicity of pinoresinol against cancer cells (https://doi.org/10.1016/j.bmcl.2017.04.087 ).

We conducted bio-guided fractionation and purification of major cardiac glycosides or cardenolides to further substantiate our hypothesis. Our ongoing preliminary research and unpublished results suggest the potential of calotropin and calactin, which are major cardenolides found in Calotropis species, to have a combined effect with several commercial chemotherapeutic agents. However, these experiments were based on the key findings of this manuscript. Therefore, we would like to present our novel results to propose the potential use of plant-derived compounds in combination with chemotherapeutic agents for enhanced benefits in cancer therapy.

Because of the high IC50 values observed in cancer treatment when using the extract fraction, combining it with a chemotherapeutic drug presents a promising approach. This strategy aims to concurrently reduce the concentration of the extract and the dosage of the anticancer drug. Such combination has the potential to enhance treatment efficacy while minimizing the side effects associated with high-dosage usage. This advantage becomes particularly evident when compared to using the extract alone, especially in cases where the isolated compound is not employed.

24) Figure 1 C would be better visualized if it were presented as a heat map to show a checkerboard analysis.

Response:

Thank you for your valuable comments.

The revised manuscript now includes the representation of the combination effect in Figure 1C and in the heat map graph (Figure 1D in the revised manuscript) as well.

25) The fractional inhibitory concentration should be computed to determine if the combinatorial effect is either synergistic, potentiation, or just an additive effect.

Response:

Thank you for your comments.

In the revised manuscript, we have addressed the combination index (CI), which assesses the synergistic efficacy of a combination treatment using the Chou-Talalay method. A CI value less than 1 indicates a synergistic mechanism for the combination treatment. We found that the combination of CGEtOAc at 400 µg/mL and sorafenib at 4 mM resulted in a CI of 0.5, with an inhibition rate of approximately 60%, demonstrating a synergistic effect. However, when CGEtOAc was used at 200 µg/mL in combination with sorafenib at 4 mM, the CI was 0.5, but the inhibition rate was 40%.

Additionally, when CGEtOAc using concentrations of 200, 400, 600, and 800 µg/mL was combined with sorafenib at 8 mM (IC50), the CI was consistently less than 1. As a result, we determined that CGEtOAc at 400 µg/mL and sorafenib at 4 µM were the most suitable for the subsequent anticancer experiments. We have also included the Supporting Information in S3 Fig, which illustrates the combination index (CI) vs. fraction affected (Fa) graph of CGEtOAc in combination with sorafenib after 24 hours of incubation.

References related to the CI analysis:

https://doi.org/10.1016/j.synres.2018.04.001

https://doi.org/10.1016/j.biopha.2016.10.096

https://doi.org/10.31557/apjcp.2023.24.5.1495

26) According to Figure 1 E, CGEtOAc is still cytotoxic in normal cells tested. Hence, the IC50 of CGEtOAc should be computed and a therapeutic index determined.

Response:

Thank you for your comments.

In the revised manuscript, we have addressed the IC50 of CGEtOAc on IMR90 cells for a 24-hour period, which was approximately greater than 5,000 µg/mL. This value was notably higher than the effect observed on HepG2 cells, where the IC50 for CGEtOAc was approximately 771.00 ± 69.74 µg/mL, around 6.5 times lower.

The toxicity of CGEtOAc at concentrations of 200, 400, and 800 µg/mL was significantly higher compared to the vehicle control for both IMR-90 and HepG2 cells, though the effect was more pronounced in HepG2 cells. Furthermore, when considering the combination of CGEtOAc at 400 µg/mL and sorafenib at 4 µM, it resulted in a significant cytotoxic effect on IMR-90 cells compared to the vehicle control, although the effect was less pronounced than what was observed in HepG2 cells. Specifically, the combination of CGEtOAc at 400 µg/mL and sorafenib at 4 µM resulted in a 60% cytotoxic effect on HepG2 cells, while it induced a 20% cytotoxic effect on IMR-90 cells. It's important to note that the combination index (CI) of CGEtOAc with sorafenib in IMR-90 cells cannot be calculated

27) Migration studies should be placed as a separate figure.

Response:

Thank you for your suggestions.

The revised manuscript has been edited, and the order of the figures has been rearranged to include seven figures. The migration results were separated into Figure 2A and 2B with some modifications. This was done as a result of repeating the migration experiment to confirm the obvious migration inhibition of the combination of CGEtOAc at 400 µg/mL and sorafenib at 4 µM on HepG2 cells, in comparison to the vehicle. Additionally, we have provided the migration rate (%) in the Supporting Information (S4 Fig.).

28) The results shown in Figure 1E and 1G did not show that the gap in the vehicle and CGEtOAc-treated cells were different and hence conclusion from this is not supported. The gap made from the wound healing assay for vehicle alone is still comparable with the baseline, hence, this assay should still be optimized before using any test compounds and extracts.

Response:

Thank you for your comments.

The revised manuscript has been edited to address the details of the wound healing assay used to evaluate the anti-migration properties of CGEtOAc on HepG2 cancer cells. We repeated the experiments to validate the migration inhibition of the treatment, and the results showed some modifications. The gap or migration distance of the vehicle group was significantly reduced in a time-dependent manner, while the treatment with CGEtOAc, sorafenib, and the combination showed no change over the 0-72 hour period, suggesting an increase in the anti-proliferation and anti-migration activities in HepG2 cells. Additionally, we have provided the migration rate (%) in Supporting Information S4 Fig.

We would like to clarify to the reviewer that we optimized the gap distance at the time of initiating the scratching in all culture wells to minimize variations. To ensure consistency in gap measurements, we followed these steps:

1. HepG2 cells were seeded in a 12-well culture plate to allow cells reaching 80-90% confluency.

2. The scratch must be done gently at the center with the straight line of each well.

3. Experiments were carried out in triplicate wells and repeated at least three times for the collection of validity of mean value.

4. Gap distance was measured at the first time of the scratching to make sure no variation of the gap before treatment started.

5. The gap width of the wound was measured and recorded in three regions, with triplicate measurements of each region performed to obtain the mean value, to measure the distance between the closest points on both sides of the wound

6. The scratch widths of the control and treatment wells were measured and normalized according to their respective values at 0 hour.

7. The gap or migrated distance (%) was defined as well as the migration rate (%).

29) For all flow cytometry experiments, gating paradigms prior to the final image should be shown in the Supplementary Information.

Response:

Thank you for your suggestions.

The revised manuscript now includes the Supporting Information, S5 Raw images, which contains raw images demonstrating the gating strategy for flow cytometry of Annexin V/PI staining for CGEtOAc at 400 µg/mL, sorafenib at 4 µM, and the combination of CGEtOAc at 400 µg/mL and sorafenib at 4 µM. Additionally, we tested sorafenib at 4, 8, and 20 µM as a positive control for apoptosis. The raw images in S5 demonstrate the flow cytometry gating strategies for both scatter and singlets gating, and they also provide apoptotic bar graphs. It's worth noting that the apoptotic rates, obtained from both the scatter and singlets gating methods, showed similar values. Importantly, the apoptotic rates presented in Figure 3 and Figure 6 in the revised manuscript are based on the scatter gating method.

30) Show how many events were collected for all flow cytometry experiments.

Response:

Thank you for your comments.

We have included Supporting Information, S5 Raw images, which demonstrate the gating strategy for flow cytometry and the flow analysis settings.

31) The statement “the combination of CGEtOAc and sorafenib promoted apoptosis in HepG2 cells depending on MMP damage” is not fully supported by the results presented. All the results have shown is that the combination of CGEtOAc and sorafenib causes early apoptosis and MMP disruption but it doesn’t necessarily mean that they are directly related to each other or the events are sequential. A temporal aspect to the experiment could be introduced to determine which events come first.

Response:

Thank you for your comments.

Our manuscript suggests apoptosis activity of the extract and sorafenib treatment in HepG2 cells, which correlated with mitochondrial damage. Our study's limitation is that it does not assess the direct relationship or sequence of apoptotic events that follow mitochondrial damage. Therefore, we stated in the suggestion that “the combination of CGEtOAc and sorafenib promoted apoptosis in HepG2 cells is correlated with MMP damage” However, several studies have been reported that apoptosis pathway mediated by mitochondrion lead to activate of caspases cascade to trigger cell apoptosis. Mitochondria plays a vital role in regulation of apoptosis pathway by trigger the delivery of cytochrome c to activate the initiator caspase, eventually resulting in cell apoptosis. 

Example of references:

https://doi.org/10.1016/j.phymed.2022.154528

https://doi.org/10.1016/j.ejphar.2017.12.027

http://dx.doi.org/10.2174/1871520620666200624145217

https://doi.org/10.1002/ptr.7054

http://www.ncbi.nlm.nih.gov/pmc/articles/pmc4502999/

32) JC-1 and Hoechst 33342 imaging should be a separate Figure.

Response:

Thank you for your comments.

The revised manuscript has been edited, and the order of figures has been adjusted to include 7 figures. The JC-1 and Hoechst 33342 imaging have been separated and are now presented in Figures 4 and 5.

33) The header “The combination of CGEtOAc and sorafenib triggered HepG2 apoptosis through increasing cellular ROS levels” should be changed as this experiment only proves that early apoptosis caused by CGEtOAc and sorafenib combination is reduced when cellular ROS is blocked but it does not fully support that increased ROS happens before early apoptosis.

Response:

Your comments are appreciated.

The revised manuscript has been edited, and the heading of the results section has been changed to read: "The combination of CGEtOAc and sorafenib triggered HepG2 apoptosis associated with an increase in cellular ROS levels.”

34) In Figure 3B, despite being statistically significant, the ROS levels are already high even if it is just in the single treatment of sorafenib and CGEtOAc, hence, the combination of the two triggering an increase of ROS levels may not be necessarily true and this is just an additive effect rather than synergy.

Response:

Thank you for your comments.

We acknowledge the reviewer's comment regarding the significant increase in ROS levels with single treatments of CGEtOAc or sorafenib, compared to the 100% vehicle control. Furthermore, the combination of these treatments significantly augmented ROS levels, reaching 193% in comparison to the single treatments (with sorafenib at 175% and CGEtOAc at 175%).

It has been reported that apoptotic cell death can also occur through the initiation of various types of stress-induced damage, with ROS production being a critical stressor. Additionally, cancer cells inherently produce higher levels of ROS than normal cells. Therefore, the environmental stress in cancer cells can dramatically activate ROS generation, posing a significant risk of oxidative-induced apoptosis. Increased intracellular ROS generation is indeed crucial for inducing apoptosis with chemotherapeutic agents in various cancer cell types. ROS levels were significantly increased approximately two-fold following the anticancer treatments. (References are provided in the section below.)

References:

http://dx.doi.org/10.1016/j.biopha.2016.10.096

https://doi.org/10.3892/ijmm.2018.3807

https://doi.org/10.1038/bjc.2013.334

https://doi.org/10.3390/ijms20184407

https://doi.org/10.31557/APJCP.2023.24.5.1495

https://doi.org/10.1016/j.semcdb.2017.05.023

https://doi.org/10.7150/thno.46728

https://doi.org/10.1007/s00253-016-7930-9

Several studies have reported that treatment with extracts or compounds isolated from C. gigantea results in apoptosis due to a significant increase in oxidative ROS generation in cancer cells. For instance, the ethanol extract of the whole plant of C. gigantea induced apoptosis in cancer cells, leading to an approximately threefold increase in ROS levels and a reduction in antioxidant enzyme levels compared to the control group (https://doi.org/10.1186/s12906-019-2561-1 ). ROS generation also significantly increased in A549 cells after treatment with coroglaucigenin, which was isolated from the stems and leaves of C. gigantea (https://doi.org/10.18632/oncotarget.16454 ).

As a result, ROS associated with apoptosis in combination therapy was identified as one of the major mediators of apoptosis. The levels of ROS demonstrated an additive effect rather than a synergistic effect following the combination treatment with CGEtOAc and sorafenib. This issue is discussed in detail in the discussion section.

35) Figure legends stating that these are histograms should be changed as all data shown are in bar graphs.

Response:

Thank you for your suggestions.

The revised manuscript has been edited, and all instances of the term "histogram" have been changed to "bar graphs," as recommended.

36) The statement “Additionally, when ROS generation was inhibited, the apoptotic effect of CGEtOAc, sorafenib, and their combination was considerably reduced to baseline levels…” should be changed as they are reduced but not to baseline levels.

Response:

Thank you for your suggestions.

The revised manuscript has been edited and changed as suggested. The sentence now reads: “When ROS generation was inhibited, the apoptotic effect of CGEtOAc, sorafenib, and their combination was significantly reduced but did not return to baseline levels, in comparison to the vehicle and their single treatment groups.”

37) Indicate what $ mean in Figure 4B.

Response:

Thank you for your suggestions.

The revised manuscript has been edited. Figure 6 in the revised manuscript displays a significant comparison of the combination treatment (in the presence of NAC) with both single CGEtOAc and sorafenib treatments, represented by the symbol $.

38) Indicate concentration of NAC used.

Response:

Thank you for your comments. 

The revised manuscript has been edited to address the NAC concentration (10 mM).

39) There is no direct evidence that the downregulation of PI3K/Akt/mTOR proteins and mRNA causes early apoptosis, hence, the statement that CGEtOAc and sorafenib combination causes early apoptosis thru PI3K/Akt/mTOR inhibition should be reconstructed and conclusions be changed.

Response:

Thank you for the constructive comments.

We agree with the reviewer that a limitation of our work is the lack of direct evaluation of how the extract, combined with sorafenib, downregulates the PI3K/Akt/mTOR pathway in HepG2 cells. In the revised manuscript, we addressed this by stating, “Thus, we hypothesize that the apoptotic mechanism of CGEtOAc combined with sorafenib is associated with the downregulation of the upstream PI3K/AKT/mTOR signaling cascade pathway. Further experiments are needed to assess how CGEtOAc in combination with sorafenib downregulates the PI3K/Akt/mTOR pathway to regulate apoptosis in HepG2 cells.”

40) Explain clearly why would the cytotoxicity of each fraction from C. gigantea ST is not entirely influenced by a single component present in each fraction.

Response:

Thank you for your question and for providing feedback to us.

The fractions of C. gigantea obtained from different solvents, each with varying polarity properties, resulted in each fraction containing different phytochemicals. The presence of a variety of phytochemicals in each C. gigantea fraction, such as phenolics, triterpenoids, cardiac glycosides, and flavonoids, led to various biological activities with differing magnitudes. These variations are attributed to differences in their chemical structures, physicochemical properties, and affinities to target proteins. Therefore, the biological activities, including cytotoxicity of each fraction, were caused by a combination of chemicals within, as previously reported and discussed (Winitchaikul T et al., 2021, Sawong S et al., 2022).

41) If it is hypothesized that a cardiac glycoside would be a key component in the EA fraction of C. gigantea ST extract, this should be isolated and further tested in the battery of assays.

Response:

Thank you for your valuable suggestion.

We conducted bio-guided fractionation and purification to isolate major cardiac glycosides or cardenolides, as a means of further supporting our hypothesis. Our ongoing preliminary research and unpublished results have indicated the potential of calotropin and calactin, two major cardenolides found in Calotropis species, in combination with several commercially available chemotherapeutic agents. However, these experiments are an extension of the key results presented in this manuscript. Therefore, we are currently sharing our novel findings to propose the potential benefits of using plant-derived compounds in conjunction with chemotherapeutic agents for cancer therapy.

 

We are genuinely appreciative of the Editor and Reviewers' decision to allow us the opportunity to revise this manuscript to fully meet PLOS ONE's publication criteria. We are thankful for the time, effort, and assistance you have provided in enhancing the quality of this work. We sincerely hope that our revisions have elevated the manuscript's quality, and we are confident that the reviewers will find it suitable for publication.

Sincerely yours,

On behalf of all co-authors,

Corresponding authors

---

## [Decision Letter · Decision Letter 1]

4 Jan 2024

PONE-D-23-18216R1Combination of Ethyl acetate fraction from Calotropis gigantea stem bark and sorafenib induces apoptosis in HepG2 cellsPLOS ONE

Dear Dr. Srisawang,

Thank you for submitting your manuscript to PLOS ONE. After careful consideration, we feel that it has merit but does not fully meet PLOS ONE’s publication criteria as it currently stands. Therefore, we invite you to submit a revised version of the manuscript that addresses the points raised during the review process.

We look forward to receiving your revised manuscript.

Kind regards,

Nafees Ahemad

Academic Editor

PLOS ONE

Journal Requirements:

Reviewers' comments:

Reviewer's Responses to Questions

**Comments to the Author**

1. If the authors have adequately addressed your comments raised in a previous round of review and you feel that this manuscript is now acceptable for publication, you may indicate that here to bypass the “Comments to the Author” section, enter your conflict of interest statement in the “Confidential to Editor” section, and submit your "Accept" recommendation.

Reviewer #2: (No Response)

Reviewer #3: All comments have been addressed

2. Is the manuscript technically sound, and do the data support the conclusions?

Reviewer #2: Partly

Reviewer #3: Yes

3. Has the statistical analysis been performed appropriately and rigorously? 

Reviewer #2: Yes

Reviewer #3: Yes

4. Have the authors made all data underlying the findings in their manuscript fully available?

Reviewer #2: Yes

Reviewer #3: Yes

5. Is the manuscript presented in an intelligible fashion and written in standard English?

Reviewer #2: Yes

Reviewer #3: Yes

6. Review Comments to the Author

Reviewer #2: In this study conducted by Chaisupasakul and colleagues, the researchers explore the therapeutic potential of the ethyl acetate fraction derived from the stem bark extract of Calotropis gigantea. The focus is on its synergistic effects in combination with sorafenib against HepG2 cells. The research findings demonstrate that the combined application of the extract and sorafenib leads to early apoptosis. The study delves into potential underlying mechanisms, including the examination of factors such as reactive oxygen species (ROS) production and the inhibition of the PI3K/Akt/mTOR pathway. The manuscript has greatly improved since the first submission. However, problems with the determination of the IC50 is an issue because of the lack of data points towards the bottom of the asymptotic curve used to compute for the IC50. This would in turn, have a cascading effect on the downstream experiments presented since the assessment whether the effect is synergistic or additive would be hard to determine. Further suggestions are written below.

1) In the abstract, “The cytotoxicity of the ethyl acetate fraction of the Calotropis gigantea (L.) Dryand. (C.gigantea) stem bark extract (CGEtOAc) of has been demonstrated in many types of cancers.” should be changed to “The cytotoxicity of the ethyl acetate fraction of the Calotropis gigantea (L.) Dryand. (C.gigantea) stem bark extract (CGEtOAc) has been demonstrated in many types of cancers.”.

2) In the Materials section of the Methodology, there should be a comma after (DMEM)(Corning, USA) instead of a period.

3) Include catalog numbers of FBS, DMEM, MTT, and Lumina Forte Western HRP Substrate.

4) “The ultrasonic-assisted extraction with 95% ethanol was used to prepare the crude extract before fractionation using the liquid-liquid chromatography of water and dichloromethane, followed by the rest of the water layer and ethyl acetate.” should be “The ultrasonic-assisted extraction with 95% ethanol was used to prepare the crude extract before fractionation using liquid-liquid chromatography of water and dichloromethane, followed by the rest of the water layer and ethyl acetate.”

5) The 50 in IC50 should be a subscript.

6) Graph Prism version 9 should be changed to GraphPad Prism version 9.

7) “The scratch wounds were consistently in size in each well to avoid variations between conditions” should be “The scratch wounds were consistent in size in each well to avoid variations between conditions”

8) There should be no indentation before HepG2 cells in the Fluorescence labeling with JC-1 to evaluate mitochondrial membrane potential section of the Methodology.

9) 5-Diphenyltetrazolium Bromide should be 5-diphenyltetrazolium bromide.

10) In S2 Fig, adjust the graph such that the origin would meet at y=0 and x=-2.0.

11) Make sure that all supplementary figures are titled properly just like how you would normally title a main text figure.

12) There is a major concern on how the effectivity of CGEtOAc is and the how the CGEtOAc and sorafenib are set up. The asymptotic curve used to determine the IC50 of the extract and the compound is missing the flat slope to the right (observations with 0% inhibition), hence the accuracy of the IC50 evaluation is compromised since the formula from GraphPad would have no bottom response to be used. This could potentially cause a cascade effect since the next experiments were based off of the computed CIs with the IC50 of CGEtOAc and sorafeninb as a baseline.

13) The subtitle “CGEtOAc in combination with sorafenib inhibited the migration ability of HepG2 cells” should be CGEtOA and sorafenib, singly and in combination inhibited the migration of HepG2 cells”.

14) Comple this Figure tile in Figure 2. The migratory capacity of HepG2 cells was assessed using a 72-h.

15) Why is there no data point for 72h treatment for CGEtOAc and sorafenib combination?

16) There should be a T-test comparing the combination group with the single treatment to determine whether the combination is actually acting on the migration rate and not just the individual ability of CGEtOAc and sorafenib. An experiment with lower doses combination can be done to determine if it is truly the combination of the two that is inhibiting migration.

17) Fix S4 Fig title as it doesn’t only show 24 h of incubation and not just the combination treatment. This applies to all the Figure titles, as they should be complete and should completely describe the data presented.

18) Define MMP.

19) Figure 3 title should be The apoptotic effect of CGEtOAc, sorafenib, and their combination against HepG2 cells after exposure for 24h.

20) “As shown in Fig 5A, 400 CGEtOAc μg/mL combined with 4 μM sorafenib upregulated ROS production with the green fluorescence intensity of dichlorofluorescein compared to the vehicle control and their respective treatment groups.” should be “As shown in Fig 5A, 400 μg/mL CGEtOAc combined with 4 μM sorafenib upregulated ROS production with the green fluorescence intensity of dichlorofluorescein compared to the vehicle control and their respective treatment groups.”

21) The subtitle “The apoptotic response following treatment with a combination of CGEtOAc and sorafenib involved the inhibition of the PI3K/Akt/mTOR pathway in HepG2 cells” should be “The apoptotic response following treatment with a combination of CGEtOAc and sorafenib may involve the inhibition of the PI3K/Akt/mTOR pathway in HepG2 cells” since the blot experiments did not show whether inhibition of PI3K/Akt/mTor by CGEtoAc and sorafenib and their combination is happening dependently or independently of apoptosis.

22) “The ethyl acetate fraction contains the highest levels of cardiac glycosides and phenolics, however, triterpenoids, flavonoids, and calotropins are present in lover levels than dichloromethane and other extract fractions.” should be “The ethyl acetate fraction contains the highest levels of cardiac glycosides and phenolics, however, triterpenoids, flavonoids, and calotropins are present in lower levels than dichloromethane and other extract fractions.”

23) Make sure that all compounds that are written in the article are in lower case (i.e. lanatoside C, raddeanin A, kaemperol, etc.) since these chemical compounds are not proper nouns.

24) The statement “Inhibition of PI3K/Akt/mTOR expression was also demonstrated as a mediator of apoptosis following combination therapy.” should be rewritten as a hypothesis or a possibility as the experiments presented are not a direct evidence of its association with apoptosis.

Reviewer #3: The revised version of this manuscript from Chaisupasakul et al. has improved considerably from its previous version. However, there are some minor suggestions in the discussion section that need to be addressed before recommending it for publication.

1. The authors have mentioned the possible role of cardiac glycoside in conferring anticancer properties, and so based on their findings in HPLC-ESI-MS, the line in discussion “the cardiac glycoside composition in the ethyl acetate fraction of the C. gigantea stem bark extract may be postulated to be one of the key components that confer anticancer properties to the ethyl acetate fraction when combined with sorafenib in HepG2 cells” should be modified accordingly.

2. The present study reports exposure to CGEtOAc and sorafenib reduces the expression of PI3K/Akt/mTOR. In contrast, previous studies from Dai et al. 2018 and Wang et al. 2020 have shown reduced activation of PI3K/Akt/mTOR signaling pathways when sorafenib was given with other combinations. Does CGEtOAc has any role in the transcription of PI3K/Akt/mTOR? Are there any available studies showing transcription regulation of PI3K/Akt/mTOR from any other plant extract?

3. In the discussion, the author also should shed lights on other possible mechanism participating in the apoptotic action of the current combination therapy

7. PLOS authors have the option to publish the peer review history of their article (what does this mean?). If published, this will include your full peer review and any attached files.

Reviewer #2: No

Reviewer #3: No

---

## [Author Response · Author response to Decision Letter 1]

14 Feb 2024

Reviewers' comments:

Reviewer's Responses to Questions

Comments to the Author

1. If the authors have adequately addressed your comments raised in a previous round of review and you feel that this manuscript is now acceptable for publication, you may indicate that here to bypass the “Comments to the Author” section, enter your conflict of interest statement in the “Confidential to Editor” section, and submit your "Accept" recommendation.

Reviewer #2: (No Response)

Reviewer #3: All comments have been addressed

2. Is the manuscript technically sound, and do the data support the conclusions?

Reviewer #2: Partly

Reviewer #3: Yes

3. Has the statistical analysis been performed appropriately and rigorously?

Reviewer #2: Yes

Reviewer #3: Yes

4. Have the authors made all data underlying the findings in their manuscript fully available?

Reviewer #2: Yes

Reviewer #3: Yes

 

5. Is the manuscript presented in an intelligible fashion and written in standard English?

Reviewer #2: Yes

Reviewer #3: Yes

6. Review Comments to the Author

Reviewer #2:

In this study conducted by Chaisupasakul and colleagues, the researchers explore the therapeutic potential of the ethyl acetate fraction derived from the stem bark extract of Calotropis gigantea. The focus is on its synergistic effects in combination with sorafenib against HepG2 cells. The research findings demonstrate that the combined application of the extract and sorafenib leads to early apoptosis. The study delves into potential underlying mechanisms, including the examination of factors such as reactive oxygen species (ROS) production and the inhibition of the PI3K/Akt/mTOR pathway. The manuscript has greatly improved since the first submission.

However, problems with the determination of the IC50 is an issue because of the lack of data points towards the bottom of the asymptotic curve used to compute for the IC50. This would in turn, have a cascading effect on the downstream experiments presented since the assessment whether the effect is synergistic or additive would be hard to determine. Further suggestions are written below.

1) In the abstract, “The cytotoxicity of the ethyl acetate fraction of the Calotropis gigantea (L.) Dryand. (C.gigantea) stem bark extract (CGEtOAc) of has been demonstrated in many types of cancers.” should be changed to “The cytotoxicity of the ethyl acetate fraction of the Calotropis gigantea (L.) Dryand. (C.gigantea) stem bark extract (CGEtOAc) has been demonstrated in many types of cancers.”.

Response: 

We appreciate the positive comments from the reviewer and thank you for these suggestions.

The revised manuscript has been edited as suggested.

2) In the Materials section of the Methodology, there should be a comma after (DMEM)(Corning, USA) instead of a period.

3) Include catalog numbers of FBS, DMEM, MTT, and Lumina Forte Western HRP Substrate.

Response: 

Thank you for these suggestions.

The revised manuscript has been edited as suggested.

 

4) “The ultrasonic-assisted extraction with 95% ethanol was used to prepare the crude extract before fractionation using the liquid-liquid chromatography of water and dichloromethane, followed by the rest of the water layer and ethyl acetate.” should be “The ultrasonic-assisted extraction with 95% ethanol was used to prepare the crude extract before fractionation using liquid-liquid chromatography of water and dichloromethane, followed by the rest of the water layer and ethyl acetate.”

Response: 

Thank you for these suggestions.

The revised manuscript has been edited as suggested.

5) The 50 in IC50 should be a subscript.

Response: 

Thank you for these suggestions.

The revised manuscript has been edited as suggested.

6) Graph Prism version 9 should be changed to GraphPad Prism version 9.

Response: 

Thank you for these suggestions.

The revised manuscript has been edited as suggested.

7) “The scratch wounds were consistently in size in each well to avoid variations between conditions” should be “The scratch wounds were consistent in size in each well to avoid variations between conditions”

Response: 

Thank you for these suggestions.

The revised manuscript has been edited as suggested.

8) There should be no indentation before HepG2 cells in the Fluorescence labeling with JC-1 to evaluate mitochondrial membrane potential section of the Methodology.

Response: 

Thank you for these suggestions.

The revised manuscript has been edited as suggested.

9) 5-Diphenyltetrazolium Bromide should be 5-diphenyltetrazolium bromide.

Response: 

Thank you for these suggestions.

The revised manuscript has been edited as suggested.

 

10) In S2 Fig, adjust the graph such that the origin would meet at y=0 and x=-2.0.

Response: 

Thank you for these suggestions.

The S2 Fig. has been revised as suggested and is shown below.

S2 Fig.-revised

S2 Fig. The IC50 curves for (A) CGEtOAc and (B) sorafenib in HepG2 cells, and (C) CGEtOAc in IMR-90 cells for 24 h of incubation.

 

11) Make sure that all supplementary figures are titled properly just like how you would normally title a main text figure.

Response: 

Thank you for these suggestions.

The revised manuscript and all figures have been edited as suggested.

12) There is a major concern on how the effectivity of CGEtOAc is and the how the CGEtOAc and sorafenib are set up. The asymptotic curve used to determine the IC50 of the extract and the compound is missing the flat slope to the right (observations with 0% inhibition), hence the accuracy of the IC50 evaluation is compromised since the formula from GraphPad would have no bottom response to be used. This could potentially cause a cascade effect since the next experiments were based off of the computed CIs with the IC50 of CGEtOAc and sorafenib as a baseline.

Response: 

Thank you for these suggestions. We appreciate the reviewer’s insights.

We have revised cell viability test to enhance the accuracy and reliability of IC50 value (Figure 1-revised and S2 Fig-revised with correction of Y-axis name to “% cell viability”). The revised manuscript has been edited as follows. 

CGEtOAc was tested at concentrations ranging from 0 to 2,000 µg/mL, while sorafenib concentrations ranged from 0 to 40 µM, as illustrated in Figure 1 and S2 Fig. The cytotoxic effect revealed that cell viability decreased to approximately 18% at CGEtOAc 2,000 µg/mL, with an IC50 value of 707.87±49.05 µg/mL. The IC50 of CGEtOAc from 0 to 1,000 µg/mL was 771.00 ± 69.74 µg/mL. Additionally, for sorafenib concentrations ranging from 0 to 40 µM, which also exhibited a decrease in cell viability to approximately 10%, the IC50 was 8.65±0.33 µM, and at 0 to 20 µM, it was 8.64 ± 0.32 µM, demonstrating a negligible difference.

However, concerning the reviewer’s concern about the missing the flat slope to the right of the asymptotic curve used to determine the IC50 of the extract and sorafenib, it’s important to note that the potentially cascade effect on the next experiments was not affected. This is because the cytotoxic effect of CGEtOAc and sorafenib revealed a decrease in cell viability to less than 10-20%. Additionally, in the consequence experiments aimed at evaluating the combination effect, we utilized sub-IC50 concentrations and those around IC50 for both CGEtOAc (at 200,400, 600, and 800 µg/mL) and sorafenib (at 2, 4, and 8 µM). The objective was to employ concentrations as low as possible to generate the combination effect of CGEtOAc and sorafenib.

Additionally, the absence of a flat slope reaching 0% cell viability was due to the limited solubility of CGEtOAc extracts at doses higher than 2,000 µg/mL, which we preliminarily explored. Therefore, the present study restricted the maximum dose of CGEtOAc to 2,000 µg/mL. Concerning the IC50 of sorafenib, we consistently obtained IC50 values (8.65±0.33 µM for concentrations from 0 to 40 µM) comparable to those reported in other publication (https://doi.org/10.21873/invivo.13190, https://doi.org/10.2147/DDDT.S344750 ). Consequently, the combination effect of CGEtOAc and sorafenib, as demonstrated in Figure 1C, where CGEtOAc was at 400 µg/mL and sorafenib at 4 µM, was deemed suitable for the subsequent experiments and potential future applications in treatment.

Figure 1-revised is shown below.

  

Fig 1. Cytotoxicity of the ethyl acetate fraction of C. gigantea stem bark extracts (CGEtOAc) and sorafenib against HepG2 cells. The 3-(4, 5-Dimethylthiazol-2-yl)-2, 5-diphenyltetrazolium bromide (MTT) technique was used to assess the cell viability after being treated for 24 h with (A) CGEtOAc, (B) sorafenib, and (C) their combination, demonstrated in the bar graph and (D) heat map analysis. (E) CGEtOAc and (F) the combination of CGEtOAc and sorafenib treatment on IMR-90 cells. Cells treated with 0.8% dimethyl sulfoxide (DMSO) represented the vehicle control. The significant differences in data, presented as the mean ± standard deviation (SD) from at least three different experiments, were investigated with a one-way analysis of variance (ANOVA) using Tukey's honestly significant difference (HSD) test: *; p < 0.05 vs the vehicle control, a; p < 0.05 vs a single CGEtOAc treatment at their own dose, b; p < 0.05 vs a single sorafenib treatment at their own dose.

13) The subtitle “CGEtOAc in combination with sorafenib inhibited the migration ability of HepG2 cells” should be CGEtOA and sorafenib, singly and in combination inhibited the migration of HepG2 cells”.

Response: 

Thank you for these suggestions.

The revised manuscript has been edited as suggested.

14) Comple this Figure tile in Figure 2. The migratory capacity of HepG2 cells was assessed using a 72-h.

Response: 

Thank you for these suggestions.

The revised manuscript has been edited, and the figure title in Figure 2 has been completed as followed:

“Fig 2. The migratory capacity of HepG2 cells was assessed through a 0-72-h incubation with CGEtOAc at 400 µg/mL and sorafenib at 4 µM, both singly and in combination.”

15) Why is there no data point for 72h treatment for CGEtOAc and sorafenib combination?

Response: 

We appreciate the reviewer’s suggestions.

In acknowledgment of your valuable question, we have incorporated this issue into the Results section as follows:

 

“The migration data for the combination of CGEtOAc and sorafenib at 72 h disappeared due to the complete loss of cell viability. Cells detached from the plate, rendering it impossible to measure the gap between cell scratches, attributed to the toxic effects of the combination treatment.”

16) There should be a T-test comparing the combination group with the single treatment to determine whether the combination is actually acting on the migration rate and not just the individual ability of CGEtOAc and sorafenib. An experiment with lower doses combination can be done to determine if it is truly the combination of the two that is inhibiting migration.

Response: 

We appreciate the reviewer’s suggestions.

The revised Figure 2 and S4 Fig. have been updated to include comparisons between the combination group with the single treatment, as stated as below:

“At a concentration of 400 µg/mL for CGEtOAc, 4 µM for sorafenib, and their combination, cell migration was significantly inhibited. This was demonstrated by a higher percentage of gap distance in a wound healing assay compared to each hour of incubation in the vehicle group, when 0 h set at 100% for each group (Fig 2A and 2B). The percentage of gap distance for the combination of CGEtOAc and sorafenib in each incubation period did not differ from the single treatments. However, the migration data for the combination of CGEtOAc and sorafenib at 72 h disappeared due to the complete loss of cell viability. Cells detached from the plate, rendering it impossible to measure the gap between cell scratches, attributed to the toxic effects of the combination treatment.

The migration rate, as shown in S4 Fig, confirmed a significant inhibition of HepG2 cell migration by CGEtOAc at 400 µg/mL, sorafenib at 4 µM, and their combination compared to each hour of incubation in the vehicle group. The migration rate for the combination of CGEtOAc and sorafenib in each incubation period did not differ from the single treatments.

Our findings indicate that CGEtOAc, sorafenib, and their combination upregulated the anti-proliferative and anti-migratory activities in HepG2 cells.”

Notably, suppression of migration was observed within 24 hours after incubation with CGEtOAc at 400 µg/mL, sorafenib at 4 µM, and their combination, with no significant difference among the 24-, 48- and 72-hour incubation periods. This finding aligns with reports demonstrating that the inhibition of migration remains consistent across different incubation times at a particular concentration of the agent that significant exhibited cytotoxic effects on cancer cells (https://doi.org/10.21873/anticanres.15241 , https://doi.org/10.1093/gastro/goaa072 ).

In addition, we found that the suppression of migration by the combination of CGEtOAc and sorafenib did not differ from that observed with their respective single treatments. However, cell viability showed a greater reduction in the combination group compared to each single treatment. Our findings are consistent with report demonstrating that the combination of the chemotherapeutic agent docetaxel and plant extract had a similar inhibitory effect on migration compared to single treatments, as evaluated by measuring gap width in breast cancer cells at every incubation time (https://doi.org/10.1155/2021/5517944 ). These findings suggest that cell migration was inhibited prior to cell death. After the first 24 hours post-treatment with both lethal and non-lethal doses of radiation and chemotherapy drug paclitaxel, migration of glioblastomas was inhibited prior to cell death. This suggestion proposed that cell migration should be a therapeutic target in anti-metastasis/anti-invasion strategies for improve cancer therapeutic outcomes (https://doi.org/10.1016%2Fj.bbrep.2021.101071 ).

Thus, our manuscript suggests that each single treatment, CGEtOAc and sorafenib, possesses migration inhibition properties, which may contribute to enhancing the synergistic effect of the combination treatment on anticancer efficiency.

The study evaluated the effects of CGEtOAc at 400 µg/mL, sorafenib at 4 µM, and their combination on the inhibition of cell migration. The results from both MTT and combination index analysis indicated that the combination of CGEtOAc at 400 µg/mL with sorafenib at 4 µM exhibited a combination index (CI) value of 0.5, correlated with an inhibition rate of approximately 60%, suggestive of a synergistic effect. However, combinations with lower concentrations, such as CGEtOAc at 200 µg/mL and sorafenib at 4 mM also yielded a CI of 0.5, albeit with a lesser inhibition rate of 40%. Therefore, we concluded that CGEtOAc at 400 µg/mL and sorafenib at 4 µM were selected for the further experimentation.

Below are the revised versions of Figure 2 and S4 Fig.

 

Figure 2-revised

Fig 2. The migratory capacity of HepG2 cells was assessed through a 0-72-h incubation with CGEtOAc at 400 µg/mL and sorafenib at 4 µM, both singly and in combination. (A) Wound healing images were captured with a magnification bar of 500 µm. (B) The percentage of the gap was depicted as a bar graph. Cells treated with 0.8% DMSO represented the vehicle control. The significant differences in data, presented as the mean ± SD from at least three different experiments were investigated with a one-way ANOVA using Tukey's HSD test: *; p < 0.05 vs the 0 h of the vehicle control, a; p < 0.05 compared to 24 h of incubation in the vehicle group, b; p < 0.05 compared to 48 h of incubation in the vehicle group, and c; p < 0.05 compared to 72 h of incubation in the vehicle group, with 0 hours set at 100% for each group. 

S4 Fig-revised

S4 Fig. The migration rate of HepG2 cells treated with CGEtOAc at 400 µg/mL and sorafenib at 4 µM, both singly and in combination, was evaluated using a wound healing assay after 0-72 h of incubation and compared to the vehicle group. Cells treated with 0.8% DMSO represented the vehicle control. The significant differences in data, presented as the mean ± SD from at least three different experiments were investigated with a one-way ANOVA using Tukey's HSD test: a; p < 0.05 compared to 24 h of incubation in the vehicle group, b; p < 0.05 compared to 48 h of incubation in the vehicle group, and c; p < 0.05 compared to 72 h of incubation in the vehicle group.

17) Fix S4 Fig title as it doesn’t only show 24 h of incubation and not just the combination treatment. This applies to all the Figure titles, as they should be complete and should completely describe the data presented.

Response: 

We appreciate the reviewer’s suggestions.

This title has been revised and changed to 

“S4 Fig. The migration rate of HepG2 cells treated with CGEtOAc at 400 µg/mL and sorafenib at 4 µM, both singly and in combination, was evaluated using a wound healing assay after 0-72 h of incubation and compared to the vehicle group.”

 

18) Define MMP.

Response: 

We appreciate the reviewer’s suggestions.

The revised manuscript has been edited to address the issue of “mitochondrial membrane potential (MMP)”, as suggested.

19) Figure 3 title should be the apoptotic effect of CGEtOAc, sorafenib, and their combination against HepG2 cells after exposure for 24h.

Response: 

Thank you for these suggestions.

The revised manuscript has been edited as suggested.

20) “As shown in Fig 5A, 400 CGEtOAc μg/mL combined with 4 μM sorafenib upregulated ROS production with the green fluorescence intensity of dichlorofluorescein compared to the vehicle control and their respective treatment groups.” should be “As shown in Fig 5A, 400 μg/mL CGEtOAc combined with 4 μM sorafenib upregulated ROS production with the green fluorescence intensity of dichlorofluorescein compared to the vehicle control and their respective treatment groups.”

Response: 

Thank you for these suggestions.

The revised manuscript has been edited as suggested.

21) The subtitle “The apoptotic response following treatment with a combination of CGEtOAc and sorafenib involved the inhibition of the PI3K/Akt/mTOR pathway in HepG2 cells” should be “The apoptotic response following treatment with a combination of CGEtOAc and sorafenib may involve the inhibition of the PI3K/Akt/mTOR pathway in HepG2 cells” since the blot experiments did not show whether inhibition of PI3K/Akt/mTor by CGEtoAc and sorafenib and their combination is happening dependently or independently of apoptosis.

Response: 

Thank you for these suggestions.

The revised manuscript has been edited as suggested.

22) “The ethyl acetate fraction contains the highest levels of cardiac glycosides and phenolics, however, triterpenoids, flavonoids, and calotropins are present in lover levels than dichloromethane and other extract fractions.” should be “The ethyl acetate fraction contains the highest levels of cardiac glycosides and phenolics, however, triterpenoids, flavonoids, and calotropins are present in lower levels than dichloromethane and other extract fractions.”

Response: 

Thank you for these suggestions.

The revised manuscript has been edited as suggested. 

23) Make sure that all compounds that are written in the article are in lower case (i.e. lanatoside C, raddeanin A, kaemperol, etc.) since these chemical compounds are not proper nouns.

Response: 

We appreciate the reviewer's suggestions.

The revised manuscript has been carefully reviewed and edited as suggested.

24) The statement “Inhibition of PI3K/Akt/mTOR expression was also demonstrated as a mediator of apoptosis following combination therapy.” should be rewritten as a hypothesis or a possibility as the experiments presented are not a direct evidence of its association with apoptosis.

Response: 

Thank you for these suggestions.

The revised manuscript has been edited as follows, incorporating your recommendations.

“Inhibition of PI3K/Akt/mTOR expression also supports the hypothesis that the apoptotic mechanism of CGEtOAc, when combined with sorafenib, may be associated with the downregulation of the upstream PI3K/AKT/mTOR signaling cascade pathway. However, the experiments presented did not provide sufficient evidence of this inhibition in a direct association with the apoptotic effects following the treatment.”

We would like to inform you that our manuscript's title has been edited using “ethyl acetate” to replace “Ethyl acetate”, thus the title is “Combination of ethyl acetate fraction from Calotropis gigantea stem bark and sorafenib induces apoptosis in HepG2 cells".

In conclusion, we sincerely thank the editor of the PLOS ONE and the reviewers for their insightful comments and suggestions. We appreciate the time and effort you have dedicated to assisting our work to be suitable for publication. We have carefully noted all your suggestions and comments and have addressed them, as they are beneficial for strengthening this study.

Sincerely yours,

Corresponding authors

 

Reviewer #3:

The revised version of this manuscript from Chaisupasakul et al. has improved considerably from its previous version. However, there are some minor suggestions in the discussion section that need to be addressed before recommending it for publication.

1. The authors have mentioned the possible role of cardiac glycoside in conferring anticancer properties, and so based on their findings in HPLC-ESI-MS, the line in discussion “the cardiac glycoside composition in the ethyl acetate fraction of the C. gigantea stem bark extract may be postulated to be one of the key components that confer anticancer properties to the ethyl acetate fraction when combined with sorafenib in HepG2 cells” should be modified accordingly.

Response: 

We appreciate the reviewer's suggestions.

The revised manuscript has been carefully reviewed and edited incorporating the suggested changes as follows:

“Consequently, based on the findings in HPLC-EIS-MS, it may be postulated that the cardiac glycoside composition in the ethyl acetate fraction of the C. gigantea stem bark extract is one of the key components that confer anticancer properties to the ethyl acetate fraction when combined with sorafenib in HepG2 cells.”

2. The present study reports exposure to CGEtOAc and sorafenib reduces the expression of PI3K/Akt/mTOR. In contrast, previous studies from Dai et al. 2018 and Wang et al. 2020 have shown reduced activation of PI3K/Akt/mTOR signaling pathways when sorafenib was given with other combinations. Does CGEtOAc has any role in the transcription of PI3K/Akt/mTOR? Are there any available studies showing transcription regulation of PI3K/Akt/mTOR from any other plant extract?

Response: 

We acknowledge the reviewer’s comments regarding the reduced activation of PI3K/Akt/mTOR signaling pathways instead of the transcription of protein expression following various treatments with plant extracts, sorafenib, both individually and in combination. However, it is worth noting that several reports have demonstrated the effects of plant extracts, sorafenib, and their combination on the transcription of protein signaling pathway expression in cancer cells. However, there have been no reports on the effect of C. gigantea extracts, either alone or in combination with sorafenib, on the inhibition of PI3K/Akt/mTOR expression in cancer cells. The following publications serve as examples of C. procera (the family Apocynaceae) extracts and cardiac glycosides in the inhibition of the expression of the PI3K/AKT/mTOR pathway in cancer cells.

 

The leaves of C. procera extracts exhibited inhibition on AKT/mTOR expression, inducing apoptosis in cancer cells (https://doi.org/10.1016/j.heliyon.2023.e16706 ). Cardiac glycosides, peruvoside, inhibited the expression of PI3K and mTOR in cancer cells (https://doi.org/10.1016/j.lfs.2019.117147 ). Cardiac glycoside ouabain exhibited inhibition of the expression of mTOR and p-AKT in cancer cells (https://doi.org/10.3892/mmr.2018.8587 ). Lanatoside C, the cardiac glycoside, downregulated gene expression of AKT, PI3K, and mTOR in cancer cells (https://doi.org/10.3390/biom9120792 ). Digoxin decreased PI3K and p-Akt protein expression levels in cancer cells (https://doi.org/10.1016/j.redox.2019.101131 ). Ouabain significantly decreased the levels of PI3K and p-AKT(Thr308) in cancer cells (https://doi.org/10.21873/anticanres.15241 ).

Thus, we have addressed the following points in the discussion:

“Inhibition of PI3K/Akt/mTOR expression also supports the hypothesis that the apoptotic mechanism of CGEtOAc, when combined with sorafenib, may be associated with the downregulation of the upstream PI3K/AKT/mTOR signaling cascade pathway. However, the experiments presented did not provide sufficient evidence of this inhibition in direct association with the apoptotic effects following the treatment. Furthermore, consistent with our findings, there have been reports on the effect of C. procera extracts and cardiac glycosides in the inhibition of the expression of the PI3K/AKT/mTOR pathway in correlation to induce apoptosis cancer cells (Ref: ).”

The following publications demonstrate examples of reports revealing the effects of plant extracts on the inhibition of the expression of the PI3K/AKT/mTOR pathway in cancer cells.

Matrine (MAT), a compound extracted from Sophora flavescens Aiton, exhibited anticancer against breast cancer cells in a 24-h incubation by inhibiting the expression of PI3K/AKT pathway (https://doi.org/10.1038%2Fs41598-023-39655-9 ).

The anticancer activity against renal cell carcinoma of gypenosides of Gynostemma pentaphylla (Thunb.) Makino was demonstrated through the downregulation of mRNA and protein expressions of the AKT/P-AKT-mTOR-P-mTOR pathway (https://doi.org/10.1016/j.jep.2021.113907 ).

Bupleurum, the dried root of Bupleurum chinensis DC. downregulates both the genes and protein expressions of PI3K, Akt, mTOR and phosphorylated proteins P-PI3K, P-Akt, P-MTOR (https://doi.org/10.1016/j.jep.2021.114742 ), along with other publications including Celastrus orbiculatus extracts ((http://dx.doi.org/10.2174/1871520619666190731162722), Scutellaria barbata D.Don (https://doi.org/10.3892/or.2017.5892 ).

 

Additionally, anticancer drugs have been reported to exhibit potential anticancer effects through the inhibition of protein expressions in PI3K/Akt/mTOR signaling pathways. Sorafenib has been shown to suppress PI3K expression (https://doi.org/10.3389/fonc.2022.852095 , https://doi.org/10.3892%2Fol.2018.8536, http://www.ncbi.nlm.nih.gov/pmc/articles/pmc6789287/, ), decrease the expression of the PI3K/Akt pathway (doi: 10.1186/s12964-023-01355-2, https://doi.org/10.18632/oncotarget.9168 ), suppress AKT expression downstream effectors including 4E-BP1 and eIF4E (DOI: 10.1097/CAD.0000000000001056), and downregulate the PI3K/AKT/mTOR pathway (PMID: 25778319).

We also addressed in the discussion that there have been reports indicating that sorafenib, in combination with other plant extracts, exhibited suppression of PI3K/Akt/mTOR expression in cancer cells.

“Furthermore, osthole (coumarins), sorafenib, and the combination of both compounds suppressed PI3K and Raf kinases, resulting in cytotoxic effects in cancer cells (https://doi.org/10.3390/molecules25215192 ). Additionally, sorafenib in combination with capsaicin downregulated the expression levels of Akt, mTOR and p70S6K in LM3 human hepatocellular carcinoma cells, thereby inducing apoptosis (https://doi.org/10.3892/or.2018.6754).”

3. In the discussion, the author also should shed lights on other possible mechanism participating in the apoptotic action of the current combination therapy

Response: 

Thank you for these suggestions, which contribute to strengthening the manuscript's correlation between results and the potential mechanisms underlying the anticancer effect of the combination of CGEtOAc and sorafenib.

This manuscript has provided a discussion into the possible mechanism of HepG2 cells exposed to a combination of CGEtOAc and sorafenib, leading to activation of apoptosis. ROS generation was found to be predominantly responsible for the apoptosis-inducing effects of the combined therapy; however, ROS is not the sole mechanism participating in the apoptotic action. Inhibition of PI3K/Akt/mTOR expression also supports the hypothesis that the apoptotic mechanism of CGEtOAc, when combined with sorafenib, may be associated with the downregulation of the upstream PI3K/AKT/mTOR signaling cascade pathway. Other mechanisms may work in conjunction to generate this treatment efficiency.

We also discussed that ROS production has been reported as a regulator of PI3K/Akt/mTOR expression, which results in cancer cell apoptosis. Furthermore, we discussed sorafenib's mechanism in triggering apoptosis in cancer cells. Moreover, this manuscript addressed the possible mechanism of a combination of sorafenib with anticancer agents, which promotes cancer cell apoptosis.

 

Regarding the review’s comment that provides strength to the potential discussion of this study, the revised manuscript has addressed the possible mechanism of the anticancer effect of a combination of C. gigantea extracts and sorafenib as follows:

“Notably, several studies have reported the apoptotic effect of cardiac glycosides, with this compound identified as one of the major components in C. gigantea extracts that exhibit anticancer activity. This effect is achieved through the mechanism of inhibiting Na+/K+ ATPase, leading to an increase in intracellular Ca2+ via the Na+/Ca2+ exchanger https://doi.org/10.1007/s10549-005-9053-3 , https://doi.org/10.1371/journal.pone.0287769 . Cardenolides from C. gigantea exhibited potent anticancer effects against breast cancers by inhibiting Na+/K+ ATPase, resulting in an increase in intracellular Ca2+ via the Na+/Ca2+ exchanger-dependent manner, ultimately leading to apoptosis https://doi.org/10.1021/acs.jnatprod.0c00423 . Cardenolide UNBS1450, having lower affinity for the α-2 Na+/K+ ATPase than other cardenolides, induced downregulation of myeloid cell leukemia-1, a member of the anti-apoptotic protein family downstream of Na+/K+ ATPase activity, contributing to apoptosis in cancer cells https://doi.org/10.1038/cddis.2015.134 . Various signaling pathways downstream of Na+/K+ ATPase inhibition has been demonstrated to contribute to the inhibition of cancer cell proliferation. Oleandrin, a cardiac glycoside extracted from the leaves of Neriaum oleander, induced accumulation of Ca2+, leading to ER stress-mediated cytotoxic effects, enhancing immunogenic cell death in both in vitro and in vivo cancer models https://doi.org/10.1038/s41419-021-03605-y . Ascleposide, a natural cardenolide isolated from Reevesia formosana, downregulated the oncoprotein c‐Myc protein and inhibited the phosphorylation of tumor suppressor protein Rb, resulting in the inhibition of cancer cell cycle progression and blocking cell proliferation. This effect was attributed to an increase of α‐tubulin acetylation, which contributed to the inhibition of Na+/K+ ATPase activity by enhancing the endocytosis of this protein in cancer cells through activation of the mitogen‐activated protein kinases (MAPKs) pathway https://doi.org/10.1002/pros.23944 .

Additionally, toxicarioside G, a cardenolide isolated from C. gigantea, exhibited another mechanism of anticancer activity by inhibiting cancer cell viability and proliferation. It enhanced Yes1 associated transcriptional regulator (YAP) dephosphorylation and nuclear localization, along with downstream target gene expression https://doi.org/10.3892/or.2021.8175 . Calotropin, a cardiac glycoside derived from C. gigantea, demonstrates an inhibitory effect on the regulation of metabolic reprogramming in cancer, specifically targeting aerobic glycolysis (the Warburg effect). This leads to activation of cell cycle arrest and inhibition metastasis in cancer cells, ultimately contributing to activation of apoptosis https://doi.org/10.1016/j.jobcr.2023.09.002. Another cardiac glycoside, 3′-epi-12β-hydroxyfroside, isolated from the roots of C. gigantea, induced autophagy, leading to enhanced apoptosis. This effect was mediated by downregulation of the heat shock protein 90 (Hsp90)-regulated Akt/mTOR pathway in lung cancer cells https://doi.org/10.7150/thno.23304. Lanatoside C, a cardiac glycoside, has been shown to induce apoptosis in cancer cells by suppressing the Janus kinase (JAK)/signal transducer and activator of transcription (STAT)/cytokine signaling (SOCS) (JAK2/STAT6/SOCS2) pathway http://www.ncbi.nlm.nih.gov/pmc/articles/pmc8387865/ , DOI: 10.3892/ol.2021.13001.”

“In addition to the possible mechanism proposed in this study for CGEtOAc, combined with sorafenib-induced apoptosis in cancer cells, which involves increased ROS accumulation and inhibition of PI3K/Akt/mTOR expression, the apoptotic effects resulting from the combination treatment of compounds found in C. gigantea extracts and anticancer agents may be attributed to several mechanisms. For instance, the synergistic effect of sorafenib in combination with the Na+/K+ ATPase inhibitor berbamine, an alkaloid isolated from Berberis amurensis, and cardiac glycosides like ouabain, is proposed to be contributed by the potentiation of epidermal growth factor receptor (EGFR)-mediated ERK1/2 and p38MAPK activation by berbamine https://doi.org/10.1111/bph.15616 . Similarly, the combination of digitoxin and sorafenib has been reported to suppress p-ERK, hypoxia inducible factor (HIF)-1α, HIF-2α, and VEGF expression, contributing to apoptosis in cancer cells https://doi.org/10.3892/mmr.2016.6096 . Furthermore, the combination of cardenolide derivative, AMANTADIG (3β-[2-(1-amantadine)-1-on-ethylamine]-digitoxigenin) and docetaxel induced apoptosis through the inhibition of surviving protein expression in human androgen-insensitive prostate cancer cells. (https://doi.org/10.1016/j.biopha.2018.08.028 ).”

We would like to inform you that our manuscript's title has been edited using “ethyl acetate” to replace “Ethyl acetate”, thus the title is “Combination of ethyl acetate fraction from Calotropis gigantea stem bark and sorafenib induces apoptosis in HepG2 cells".

In conclusion, we sincerely thank the editor of PLOS ONE and the reviewers for their insightful comments and suggestions. We appreciate the time and effort you have dedicated to assisting our work to be suitable for publication. We have carefully noted all your suggestions and comments and have addressed them, as they are beneficial for strengthening this study.

Sincerely yours,

Corresponding authors

---

## [Editor Report · Decision Letter 2]

21 Feb 2024

Combination of ethyl acetate fraction from Calotropis gigantea stem bark and sorafenib induces apoptosis in HepG2 cells

PONE-D-23-18216R2

Dear Dr. Srisawang,

We’re pleased to inform you that your manuscript has been judged scientifically suitable for publication and will be formally accepted for publication once it meets all outstanding technical requirements.

Kind regards,

Nafees Ahemad

Academic Editor

PLOS ONE
---

## [Editor Report · Acceptance letter]

14 Mar 2024

PONE-D-23-18216R2 

PLOS ONE

Dear Dr. Srisawang, 

I'm pleased to inform you that your manuscript has been deemed suitable for publication in PLOS ONE. Congratulations! Your manuscript is now being handed over to our production team.

Kind regards, 

on behalf of

Dr. Nafees Ahemad 

Academic Editor

PLOS ONE